# Smoothed Robustness Analysis: Bridging worst- and average-case robustness analyses via smoothed analysis

**Thomás Rodrigues Crespo**                                        *thomas.crespo.23m@st.kyoto-u.ac.jp*
*Graduate School of Informatics*
*Kyoto University*

**Jun-nosuke Teramae**                                              *teramae@acs.i.kyoto-u.ac.jp*
*Graduate School of Informatics*
*Kyoto University*

**Reviewed on OpenReview:** *https://openreview.net/forum?id=BogwFMz5tU*

## Abstract

The sensitivity to adversarial attacks and noise is a significant drawback of neural networks, and understanding and certifying their robustness has attracted much attention. Studies have attempted to bridge two extreme analyses of robustness; one is the worst-case analysis, which often gives too pessimistic certification, and the other is the average-case analysis, which often fails to give a tight guarantee of robustness. Among them, *Randomized Smoothing* became prominent by certifying a worst-case region of a classifier under input noise. However, the method still suffers from several limitations, probably due to the lack of a larger underlying framework to locate it. Here, inspired by the *Smoothed Analysis* of algorithmic complexity, which bridges the worst-case and average-case analyses of algorithms, we provide a theoretical framework for robustness analyses of classifiers, which contains *Randomized Smoothing* as a special case. Using the framework, we also propose a novel robustness analysis that works even in the small noise regime and thus provides a more confident robustness certification than *Randomized Smoothing*. To validate the approach, we evaluate the robustness of fully connected and convolutional neural networks on the MNIST and CIFAR-10 datasets, respectively, and find that it indeed improves both adversarial and noise robustness.

## 1 Introduction

Deep learning has significantly advanced machine learning in various real-world tasks over the last decade. Despite its success, deep learning is not sufficiently robust to small input perturbations known as adversarial perturbations Szegedy et al. (2014) or intrinsic noise that can occur in the inputs. Besides research to improve the adversarial and noise robustness of neural networks, theoretical studies to quantify and certify robustness have also emerged to address the problem.

One extreme of robustness quantification of neural networks is adversarial robustness certification, which tries to identify a set of inputs that the model is guaranteed to classify correctly. Thus, a larger set indicates a more robust model. A major example of this is *Certified Radius*, which is the radius of a ball such that it is centered on a correctly classified input and does not contain adversarial examples. Since this analysis provides a safe region against any perturbation, it gives a worst-case robustness of the model. However, the analysis may give a too pessimistic view of the real-world performance of the model, as the *Certified Radius* is often smaller than the spread of natural noise.

The other extreme is robustness analysis using random input perturbations (Franceschi et al., 2018; Weng et al., 2019; Couellan, 2021; Tit & Furon, 2021; Anderson & Sojoudi, 2023). Since the average performance of a classifier is given by its classification probability, which is difficult to obtain, in practice one analyzes

its lower and upper bounds. Then, a higher lower bound for larger variances indicates higher robustness to noise. In contrast to *Certified Radius*, the expectation over all directions, which mitigates the effect of misclassification regions with small volume, gives a better measure of empirical performance. However, the analysis can be too optimistic since, for instance, it gives "infinite" robustness for linear classifiers in binary classification because the misclassification probability is always below 0.5 for finite noise strength.

Given the above, approaches have attempted to bridge those two extremes (Fawzi et al., 2016; Rice & Bair, 2021; Robey et al., 2022), with *Randomized Smoothing* (Cohen et al., 2019) being the most prominent. In this analysis, one obtains *Certified Radius* of a smoothed classifier, a smoothed version of a deterministic classifier given by injecting noise into the input. Namely, the robustness of the classifier is measured here by the radius of a ball around the input, within which all points are classified correctly on average. As in average-case robustness, smoothing mitigates the effect of small regions. In addition, the analysis can scale to arbitrary neural networks. However, the analysis still has limitations. For instance, its *Certified Radius* scales with the variance of the input noise, resulting in vanishing it for small variances and information loss for larger variances Mohapatra et al. (2021). It is also unclear whether the analysis can be placed in a general theoretical framework and whether we can extend it based on the framework.

Here, to address these problems, we borrow an idea from studies of algorithmic complexity. Inspired by the *Smoothed Analysis* of algorithmic complexity, which bridges the worst-case and average-case analyses of algorithms (Spielman & Teng, 2004), we provide a novel theoretical framework for the robustness analysis of classifiers. *Smoothed Analysis* aims to obtain a more faithful description of the empirical algorithmic time complexity compared to worst-case and average-case by obtaining the worst-case performance when subject to random perturbations, which is very similar to the protocol of *Randomized Smoothing*. However, applying the *Smoothed Analysis* of algorithmic complexity to the robustness analysis of neural networks is not straightforward, because one needs to identify a suitable performance metric to which the analysis is applied. For this problem, we show that a loss function satisfying a certain condition can be a suitable performance metric for the analysis in the robustness setting for classification tasks. We then show that *Randomized Smoothing* is a special case in the framework when the loss is the 0-1 loss. We call this robustness analysis framework *Smoothed Robustness Analysis, SRA*.

The above finding may imply that if we use a loss function other than the 0-1 loss as the performance metric, SRA may solve the problem of *Randomized Smoothing* that the certified region vanishes for small noise variances. Motivated by this, we propose the use of the margin loss and show that it indeed provides a *Certified Radius* that not only does not vanish for small noise variances but is more confident since the variance of the margin decreases with the noise variance until it converges to the zero-noise margin. We refer to the proposed *Certified Radius* based on the SRA with the margin loss as *Margin Smoothed Certified Radius, MSCR*.

As a proof of concept for our approach, by deriving an explicit expression for the MSCR for $K$-Lipschitz classifiers, we used it as an objective that, when maximized, is expected to induce both adversarial and noise robustness to them. Experiments were conducted on the MNIST and CIFAR-10 datasets for 1-Lipschitz fully connected and convolutional neural networks, respectively, and their robustness against adversarial perturbations, isotropic Gaussian noise, and progressive linear corruptions were quantified. We found that, under the considered settings, the maximization of MSCR improves robustness to both adversarial and random noise for a simpler dataset (MNIST), and for a more complex dataset (CIFAR-10), it leads to an increase in noise robustness while maintaining similar clean accuracy and adversarial robustness.

## 1.1 Related works

### 1.1.1 Bridging adversarial and noise robustness

To incorporate adversarial and noise robustness, various studies have explored robustness spectra locating the worst- and the average-case analyses in its opposing ends. Fawzi et al. (2016) considered the expected distance to the decision boundary over randomly sampled input with dimensions between 1 and $n_{in}$, which interpolate between the worst-case and random noise regimes. Rice & Bair (2021) proposed using the $l_q$ norm of a smooth loss with respect to a chosen measure as a robustness metric. When the uniform ball is used as

the measure, it gives the worst-case loss for $q = \infty$ and the expected loss over the ball for $q = 1$, interpolating between them through $q$. To address the lack of interpretability or impracticality of existing methods, Robey et al. (2022) introduced the probabilistically robust learning framework that interpolates between adversarial and average training by introducing the misclassification tolerance parameter $\rho$ determining the stringency of the worst loss within a feasible region. For the 0-1 loss with a uniform distribution in the input, $\rho = 0$ corresponds to adversarial training, where the loss for one sample is zero only if there is no adversarial example within a uniform ball. In contrast, when $\rho = 0.5$, the framework achieves average-case robustness, where a training sample contributes to the loss only if its classification probability is less than 0.5.

### 1.1.2 Randomized smoothing

While the above works bridge the worst and the average cases as two ends of a spectrum, *Randomized Smoothing* explicitly uses one noise and one adversarial mechanism to bridge them. *Randomized smoothing* was first proposed by Lecuyer et al. (2019) within the realms of differential privacy as PixelDP, providing radius certification of $l_1$ and $l_2$ balls of smoothed classifiers by employing the Laplace and Gaussian distributions as input noise, respectively. In their seminal work, Cohen et al. (2019) proved a tight $l_2$ certified radius for the case of Gaussian input noise and named the analysis *Randomized Smoothing*. In the analysis, a deterministic classifier that returns a 0-1 vector, with 1 in the correct class element, becomes a probabilistic one called *smoothed classifier*, which returns the probability of each class by injecting noise into the input. Then, using the probabilities of the correct and largest incorrect classes, one can obtain the certified radius of this smoothed classifier, that is, the radius of a ball around the input within which all points are correctly classified on average. In other words, it measures the worst-case robustness of an averaged, or smoothed, classifier.

Two significant properties make *Randomized Smoothing* attractive for robustness certification. First, smoothing acts as in average-case robustness, mitigating the effect of close but small misclassification regions. Second, it is scalable to arbitrary NNs: While most certification methods rely on the architecture of the neural network, such as its Lipschitz constant for Lipschitz-based *Certified Radius* (Cisse et al., 2017; Tsuzuku et al., 2018; Weng et al., 2018; Leino et al., 2021), or its activation functions as in *Certified Radius* based on convex relaxation (Weng et al., 2018; Wong & Kolter, 2018), *Randomized Smoothing* depends only on the input noise distribution and the classification probability estimation method.

### 1.1.3 Lipschitz-constrained NNs

Lipschitz-constrained neural networks have been widely studied as a promising way to improve their adversarial robustness. Although expressiveness and scalability have been major concerns of the approach, significant advances have been made very recently. For expressiveness, studies revealed that these networks can in fact ensure their expressivity even under Lipschitz constraints (Anil et al., 2019; Béthune et al., 2022). For scaling, based on the approach of Fazlyab et al. (2019), Araujo et al. (2023) and Wang & Manchester (2023) showed that constrained networks can scale to tiny-imagenet, and Losch et al. (2023) and Hu et al. (2023), using the idea of Leino et al. (2021), succeeded in showing the scaling to tiny-imagenet and even to imagenet, respectively. Recently, Delattre et al. (2024) connected *Randomized Smoothing* and Lipschitz margin using a 1-Lipschitz simplex mapping. They used the expected margin over the simplex mapping and showed improvements over usual *Randomized Smoothing* and margin certified radii. We adopted the 1-Lipschitz architecture to obtain the estimated certified radius at training time, and due to its stability by avoiding exploding and vanishing gradients.

### 1.1.4 Algorithm analysis: worst-case, average-case and smoothed

*Analysis of algorithms* is a field of computer science that focuses on comparing algorithmic performance (Cormen, 2009) to determine which algorithm performs the best under given performance metrics and constraints. One approach is the worst-case analysis, which provides the worst possible performance of an algorithm given certain constraints, such as input size. Thus, an algorithm with a good worst-case performance is guaranteed to always perform better than one with a worse worst-case. However, the worst-

case analysis can be too pessimistic, since the worst-case may comprise instances that might be rare or nonexistent in practice (Spielman & Teng, 2009; Roughgarden, 2021).

In these cases, average-case analysis provides a more realistic description by assigning a measure to the instance space and obtaining the performance metric as a weighted average (Ritter, 2000). However, in practice, average-case analysis requires overly simplified distributions, such as the uniform distribution over all possible instances, which do not correspond to the real instance distribution and yields results that strongly depend on the chosen distribution (Roughgarden, 2021).

*Smoothed Analysis* is introduced to address the limitations of worst- and average-case analyses Spielman & Teng (2004). This approach consists in incorporating randomness in the algorithm, which leads to the so-called smoothed algorithm, and then applying a worst-case analysis. Unlike average-case analysis, the distribution used in the averaging stage of *Smoothed Analysis* is preferable to have only a small variance around the state under evaluation. Therefore, the analysis is expected to avoid biases about the real underlying distribution by using small perturbations, similar to sensing noise or random faults. While *Smoothed Analysis* has inspired several studies on neural networks (Blum & Dunagan, 2002; Haghtalab et al., 2020; Robey et al., 2022), its relation to robustness analysis has not been fully explored yet.

## 2 Smoothed Robustness Analysis: Smoothed Analysis for the robustness of classifiers

In this section, we provide how the "*Smoothed Analysis* of algorithms" can be applied to the robustness analysis of neural networks and show that the derived framework indeed includes adversarial, noise, and *Randomized Smoothing* robustness as special cases.

### 2.1 Smoothed Robustness Analysis

Since *Smoothed Analysis* is given as the worst-case performance under random perturbations, to extend the framework to analyze the robustness of neural networks, we need to identify a suitable performance metric that properly measures the performance of the networks and is analytically or, at least, numerically tractable.

In the case of classification tasks, let us first choose a noise distribution $\rho(\boldsymbol{x}, \boldsymbol{\theta})$ parametrized by an input $\boldsymbol{x}$ and $\boldsymbol{\theta}$. Then, if the loss function $T(\boldsymbol{x})$ satisfies the following condition with respect to the given distribution, we will see that it can be a suitable performance metric for the smoothed analysis in the context of the robustness of the classifier.

**Condition 1** (Constant $\Gamma$). *There is some constant $\Gamma \in \mathbb{R}$ s.t. $\mathbb{E}_{\boldsymbol{x}' \sim \rho(\boldsymbol{x}, \boldsymbol{\theta})}[T(\boldsymbol{x}')] < \Gamma$ implies that $\mathcal{P}_c(\boldsymbol{x}; \rho) > 0.5$ for all $\boldsymbol{x} \in \mathbb{X}$, where $\mathcal{P}_c(\boldsymbol{x}; \rho)$ is the probability of correct classification of $\boldsymbol{x}$ under the noise distribution $\rho(\boldsymbol{x}, \boldsymbol{\theta})$.*

In addition to choosing the noise distribution $\rho(\boldsymbol{x}, \boldsymbol{\theta})$ and the loss function $T(x)$ that satisfies the above condition, let us specify a region $\Omega(\boldsymbol{x})$ around the input $\boldsymbol{x}$. Then, we can provide the theorem that establishes the *Smoothed Analysis* for the robustness of classifiers.

**Theorem 1** (Smoothed Robustness Analysis). *Let $\tilde{T}(\boldsymbol{x})$ denote the maximal average loss within the region $\Omega(\boldsymbol{x}) \subset \mathbb{X}$ around input $\boldsymbol{x}$ over the distribution $\rho(\tilde{\boldsymbol{x}}; \boldsymbol{\theta})$, parametrized by $\tilde{\boldsymbol{x}} \in \Omega(\boldsymbol{x})$ and $\boldsymbol{\theta}$, i.e.,*

$$\tilde{T}(\boldsymbol{x}) = \max_{\tilde{\boldsymbol{x}} \in \Omega(\boldsymbol{x})} \mathbb{E}_{\boldsymbol{x}' \sim \rho(\tilde{\boldsymbol{x}}; \boldsymbol{\theta})}[T(\boldsymbol{x}')]. \tag{1}$$

*If $\tilde{T}(\boldsymbol{x}) < \Gamma$, then no perturbation within $\Omega(\boldsymbol{x})$ causes a misclassification on average.*

*Proof.* The proof of the theorem directly follows from Condition 1, since the worst proportion of misclassified samples must be less than half due to condition $\tilde{T}(\boldsymbol{x}) < \Gamma$, and thus the samples are correctly classified on average. □

We call the robustness analysis of the classifier based on the above theorem *Smoothed Robustness Analysis, (SRA)* and define a classifier to be *Smoothed Robust* on input $\boldsymbol{x}$ with respect to the loss $\tilde{T}(\boldsymbol{x})$ if it satisfies the above inequality $\tilde{T}(\boldsymbol{x}) < \Gamma$ for given $\Omega(\boldsymbol{x})$ and $\rho(\tilde{\boldsymbol{x}}, \boldsymbol{\theta})$.

Depending on the choice of region $\Omega(\boldsymbol{x})$ and noise distribution $\rho(\tilde{\boldsymbol{x}}, \boldsymbol{\theta})$, this analysis yields the worst-, average-case, or a combination of both. If $\Omega(\boldsymbol{x}) = \{\boldsymbol{x}\}$, then the max operation is restricted to $\boldsymbol{x}$ itself, which gives the average-case. Conversely, if $\rho(\tilde{\boldsymbol{x}}; \boldsymbol{\theta}) = \delta(\tilde{\boldsymbol{x}})$, then the expectation operator yields the pointwise value of the loss without smoothing effect, hence it gives the worst-case. The three different cases are illustrated in Figure 1.

One of the most practical applications of this analysis is *robustness certification*. While one can use the inequality $\tilde{T}(\boldsymbol{x}) < \Gamma$ to verify the robustness of a given classifier with respect to a given region $\Omega(\boldsymbol{x})$ and distribution $\rho(\boldsymbol{x}, \boldsymbol{\theta})$, the inequality can also be used inversely to obtain a region $\Omega(\boldsymbol{x})$ (e.g., a $l_p$-ball) that satisfies the inequality for a given noise distribution. In other words, from the inequality one can find a *Certified Radius* such that inputs in the ball of the radius are always correctly classified on average under the noise distribution, that is *Smoothed Robust*. In the following, we will use this approach to show the relationship between the analysis and *Randomized Smoothing*, and to provide numerical calculations.

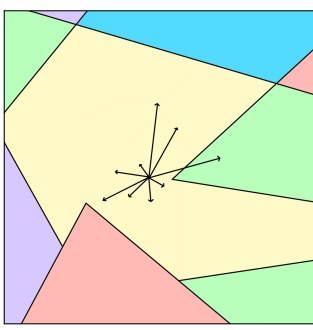 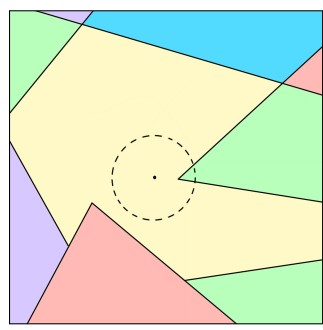 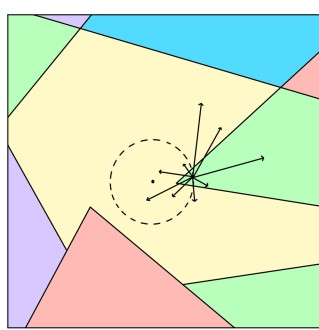

Figure 1: Illustrations of average-case (left), worst-case (center), and smoothed (right) robustness analyses. Different colors correspond to different classes according to the classifier, and the dot located in the center is the input analyzed. For average-case, if the average loss is smaller than a given constant, it is correctly classified on average. For worst-case, if the maximum of the loss inside the region is smaller than the constant, then it is safe inside the region. For the SRA, if the maximum of the average loss inside the region is smaller than the constant, then the classifier is safe inside the region on average, i.e., the smoothed classifier is safe there.

## 2.2 Randomized Smoothing as SRA with the 0-1 loss

One of the significant features of SRA is that it can yield, in theory, a family of robustness analyses by changing the loss function $T(\boldsymbol{x})$, as long as the loss satisfies Condition 1 and one can derive the constant $\Gamma$ required for the condition.

Here, we show that *Randomized Smoothing* is a special case of SRA where the loss $T(\boldsymbol{x})$ is the 0-1 loss. The 0-1 loss is defined as

$$
\mathcal{L}_{01}(\boldsymbol{x}, c) = \begin{cases} 0, & \text{if } c = \arg\max_i o_i(\boldsymbol{x}) \\ 1, & \text{otherwise} \end{cases} \tag{2}
$$

, where $o_i(\boldsymbol{x})$ is the ith component of the classification score vector and $c$ is the correct class. Then we can show that the 0-1 loss satisfies the Condition 1.

**Theorem 2.** *The 0-1 loss satisfies Condition 1 with* $\Gamma = 0.5$.

*Proof.* For an arbitrary noise distribution $\rho(\boldsymbol{x}, \boldsymbol{\theta})$, it holds that $\mathbb{E}_{\boldsymbol{x}' \sim \rho(\boldsymbol{x}, \boldsymbol{\theta})}[\mathcal{L}_{01}(\boldsymbol{x}', c)] = \mathcal{P}_{i \neq c}(\boldsymbol{x}; \rho(\cdot)) = 1 - \mathcal{P}_c(\boldsymbol{x}; \rho(\cdot))$, where $\mathcal{P}_{i \neq c}(\boldsymbol{x}; \rho(\cdot))$ and $\mathcal{P}_c(\boldsymbol{x}; \rho(\cdot))$ are the probability of incorrect and correct classification, respectively. Thus, $\mathbb{E}_{\boldsymbol{x}' \sim \rho(\boldsymbol{x}, \boldsymbol{\theta})}[\mathcal{L}_{01}(\boldsymbol{x}', c)] < 0.5$ implies $\mathcal{P}_c(\boldsymbol{x}; \rho(\cdot)) > 0.5$, which means $\Gamma = 0.5$. □

Since the theorem justifies the use of the 0-1 loss $\mathcal{L}_{01}(\boldsymbol{x}, c)$ as the loss $T(\boldsymbol{x})$ of RSA, we can use the inequality $\tilde{\mathcal{L}}_{01}(\boldsymbol{x}) = \max_{\tilde{\boldsymbol{x}} \in \Omega(\boldsymbol{x})} \mathbb{E}_{\boldsymbol{x}' \sim \rho(\tilde{\boldsymbol{x}}; \boldsymbol{\theta})}[\mathcal{L}_{01}(\boldsymbol{x}')] < 0.5$ to obtain a certified radius. However, because, as shown in the above proof, $\mathbb{E}_{\boldsymbol{x}' \sim \rho(\boldsymbol{x}, \boldsymbol{\theta})}[\mathcal{L}_{01}(\boldsymbol{x}', c)]$ is equal to the misclassification probability under given noise, i.e. misclassification probability of the smoothed classifier, this procedure means that one can obtain a certified radius based on worst-case analysis of the misclassification probability of the smoothed classifier. This is exactly the same as the general idea of *Randomized Smoothing* (Cohen et al., 2019; Lecuyer et al., 2019). For the case where the noise distribution $\rho(\boldsymbol{x}, \boldsymbol{\theta})$ is the Gaussian with variance $\sigma_x^2$ and the region $\Omega(\boldsymbol{x})$ is the $l_2$ ball around the input, the radius is given as (Cohen et al., 2019)

$$\epsilon_{01, l_2}(\boldsymbol{x}; \mathcal{N}(0, \sigma_x^2 \boldsymbol{I})) = \frac{\sigma_x}{2} \left( \Phi^{-1}(\mathcal{P}_c) - \Phi^{-1}(\max_{i \neq c} \mathcal{P}_i) \right), \tag{3}$$

where $\Phi^{-1}(\cdot)$ is the inverse of the cumulative distribution function of the standard normal distribution.

## 3 Margin-based SRA: Smoothed Robustness Analysis with the margin loss

One major drawback of *Randomized Smoothing* is that the certified radius scales with the input noise variance, as in equation 3, meaning that for small variances the certified region vanishes. However, in the literature of smoothed analysis of algorithmic complexity, the smaller the variance of the perturbation, the better, which cannot be done by *Randomized Smoothing*. Recently, Delattre et al. (2024) succeeded in eliminating this scaling factor of the input variance by first noting that, for $K$-Lipschitz classifiers, the output variance is bounded by $K^2 \sigma_x^2$, and then using the *margin* of the probability vectors instead of the probability itself. This suggests that the margin might be suitable for SRA. In this section, we will show that this is indeed the case by proposing the use of the margin loss as the loss $T(\boldsymbol{x})$ of RSA. We also derive a closed-form estimate of the certified radius under the setting of $K$-Lipschitz classifiers, which we will use in our numerical simulation given in the next section.

### 3.1 Margin Loss

The margin loss is defined as the difference between the largest output in incorrect classes and the output of the correct class,

$$\mathcal{L}_M(\boldsymbol{x}, c) = o_m(\boldsymbol{x}) - o_c(\boldsymbol{x}), \tag{4}$$

where $c$ denotes the correct class and $m$ is the largest incorrect index, $m = \arg\max_{i \neq c} o_i(\boldsymbol{x})$.

Then, for the loss, we can prove the following proposition.

**Proposition 3.** *If the output of a classifier is continuous and symmetrically distributed under an input perturbation $\rho$, the margin loss satisfies Condition 1 with $\Gamma = 0$ for the distribution.*

*Proof.* This basically follows from the inequality for the smoothed margin $\bar{\mathcal{L}}_M(\boldsymbol{x}, c; \rho(\cdot))$ of continuous and symmetrically distributed outputs,

$$\bar{\mathcal{L}}_M(\boldsymbol{x}, c; \rho(\cdot)) = \mathbb{E}_{\boldsymbol{x}' \sim \rho(\boldsymbol{x}, \boldsymbol{\theta})}[\max_{i \neq c} o_i(\boldsymbol{x}')] - \mu_c > \text{med}[\max_{i \neq c} o_i(\boldsymbol{x})] - \mu_c,$$

where $\mu_c$ is the mean correct output, med$[\cdot]$ refers to the median. The full proof is given in Appendix A.1. $\square$

This proposition justifies our use of the margin loss as $T(\boldsymbol{x})$ of RSA if the output of the classifier smoothed by the input noise is symmetric. We found this condition for the output to be adequate both empirically, as shown in Appendix D.1, and theoretically, since independent rectified Gaussian random variables satisfy the Central Limit Theorem, as shown in Appendix A.6, which means that their sum converges to a normal distribution, thus continuous and symmetrical. In the following, therefore, we assume that the output of the classifier satisfies the condition and apply the RSA based on the margin loss to them.

Let us assume that the classifier is $L_p$-Lipschitz. Then we can prove the following theorem insisting that the radius

$$\epsilon_{SM,l_p}(\boldsymbol{x};\rho(\cdot)) = -\frac{\bar{\mathcal{L}}_M(\boldsymbol{x},c;\rho(\cdot))}{\alpha_p L_p}, \tag{5}$$

is the certified radius in the sense of SRA with the margin loss, where $\bar{\mathcal{L}}_M(\boldsymbol{x},c;\rho(\cdot)) = \mathbb{E}_{\boldsymbol{x}\sim\rho(\cdot)}\left[\mathcal{L}_M(\boldsymbol{x},c)\right]$ is the margin loss smoothed by the input noise, and $\alpha_p$ is a constant depending on the norm ($\alpha_p = \sqrt{2}$ and 2 for $p = 2$ and $\infty$, respectively [1]). The theorem is given as

**Theorem 4.** *Let the region $\Omega(\boldsymbol{x})$ be the $l_2$-ball with radius $\epsilon_{SM,l_2}(\boldsymbol{x};\rho(\cdot))$. Then, $\tilde{\mathcal{L}}_M(\boldsymbol{x}) = \max_{\tilde{\boldsymbol{x}}\in\Omega(\boldsymbol{x})}\mathbb{E}_{\boldsymbol{x}'\sim\rho(\tilde{\boldsymbol{x}};\boldsymbol{\theta})}\left[\mathcal{L}_M(\boldsymbol{x}',c)\right] < 0$ for any $\delta$ s.t. $\|\delta\| < -\frac{\bar{\mathcal{L}}_M(\boldsymbol{x},c;\rho(\cdot))}{\alpha_p L_p}$.*

*Proof.* The theorem follows from two propositions. The first claims that the margin loss $\mathcal{L}_M$ of a $L_p$-Lipschitz classifier is $\alpha_p L_p$-Lipschitz, and the other says that the convolution of a $K$-Lipschitz function with a probability density function is also $K$-Lipschitz. The full proof is in Appendix A.2. $\square$

Significantly, unlike the certified radius obtained from *Randomized Smoothing* (Eq. 3), the certified radius derived here for the margin-based RSA (Eq. 5), does not vanish even in the zero-variance limit of the noise distribution $\rho(\cdot)$. Rather, it just converges to the conventional certified radius of the classifier that is not perturbed by input noise,

$$\epsilon_{M,l_p}(\boldsymbol{x}) = -\frac{\mathcal{L}_M(\boldsymbol{x},c)}{\alpha_p L_p} \tag{6}$$

This property can make the obtained certified radius more confident by avoiding the information loss due to the larger variances required to obtain meaningful certified radius under *Randomized Smoothing*. We refer to the certified radius obtained through the margin-based SRA, Eq. 5, as *Margin Smoothed Certified Radius (MSCR)*.

To proceed further and obtain a closed-form expression of the MSCR, one needs to specify the output distribution of the classifier. Assuming that the outputs can be approximated by a multidimensional normal with independent elements yields the following approximation of $\bar{\mathcal{L}}_M$. (The derivation is in the Appendix A.3),

$$\bar{\mathcal{L}}_M\left(\boldsymbol{x},c,\sigma_x^2\right) \cong \sqrt{\frac{\sigma_1^2+\sigma_2^2}{2\pi}}\exp\left(-\frac{(\mu_1-\mu_2)^2}{2(\sigma_1^2+\sigma_2^2)}\right) + \mu_2 + \frac{\mu_1-\mu_2}{2}\left(1+\mathrm{erf}\left(\frac{\mu_1-\mu_2}{\sqrt{2(\sigma_1^2+\sigma_2^2)}}\right)\right) - \mu_c. \tag{7}$$

Here, $\mu_c$, $\mu_1$ and $\mu_2$ are the mean of the correct, the largest and the second largest incorrect outputs, respectively, and $\sigma_1^2$ and $\sigma_2^2$ are the variance of the largest and the second largest incorrect outputs, respectively. By putting the expression into Eq. 5, we obtain a closed-form expression of the MSCR.

Note that the above approximation is similar to the one that Zhen et al. (2021) used to obtain the approximation of the probability of the correct classification,

$$\mathcal{P}_c\left(\boldsymbol{x},\sigma_x^2\right) \cong \frac{1}{2}\left[1+\mathrm{erf}\left(\frac{\mu_c-\mu_1}{\sqrt{2(\sigma_c^2+\sigma_1^2)}}\right)\right], \tag{8}$$

which gives a lower bound on the certified radius of the *Randomized Smoothing*,

$$\epsilon_{01,l_2} = \frac{\sigma_x}{2}\left(\Phi^{-1}(\mathcal{P}_c) - \Phi^{-1}(\max_{i\neq c}\mathcal{P}_i)\right) \geq \frac{\sigma_x}{2}\left(\Phi^{-1}(\mathcal{P}_c) - \Phi^{-1}(1-\mathcal{P}_c)\right)$$

$$= \sigma_x\Phi^{-1}(\mathcal{P}_c) \cong \sigma_x\Phi^{-1}\left(\Phi\left(\frac{\mu_c-\mu_1}{\sqrt{\sigma_c^2+\sigma_1^2}}\right)\right) = \sigma_x\frac{\mu_c-\mu_1}{\sqrt{\sigma_c^2+\sigma_1^2}}. \tag{9}$$

---

[1] We prove on Proposition 5 in Appendix A.2 that $\alpha_p = 2^{\frac{1}{q}}$, where $\frac{1}{q}+\frac{1}{p}=1$, for $0 < p,q \leq +\infty$

## 4 Proof of Concept

The properties of MSCR mentioned above may suggest that if one uses this certified radius, Eq. 5, as an objective function one can improve both adversarial and noise robustness compared to the certified radius of *Randomized Smoothing*, Eq. 3, and the conventional certified radius, Eq. 6. Here, to show that this is indeed the case, we conduct numerical experiments as a proof of concept of the proposed margin-based SRA.

### 4.1 Objective functions

Direct maximization of a certified radius without any bounds degrades the clean accuracy of classifiers, due to the trade-off between robustness and accuracy. To avoid this problem, instead of directly maximizing the certified radius, we introduce a hyperparameter $d$ that controls the maximal contribution of the certified radius and try to minimize the hinge loss,

$$\ell_{\cdot,l_2}\left(\boldsymbol{x},c,d,\sigma_x^2\right) = \mathrm{ReLU}(d - \epsilon_{\cdot,l_2}\left(\boldsymbol{x},c,\sigma_x^2\right))$$

Here, $\epsilon_{\cdot,l_2}\left(\boldsymbol{x},c,\sigma_x^2\right)$ denotes the certified radius under $\cdot$ classification loss (e.g., "01" or "$M$" for 0-1 or margin losses, respectively) under Gaussian input noise with variance $\sigma_x^2$. In our experiments, we minimized the hinge loss with MSCR,

$$\ell_{SM,l_2}\left(\boldsymbol{x},c,d,\sigma_x^2\right) = \mathrm{ReLU}(d - \epsilon_{SM,l_2}\left(\boldsymbol{x},c,\sigma_x^2\right)) = \mathrm{ReLU}\left(d + \frac{\bar{\mathcal{L}}_M(\boldsymbol{x},c;\rho(\boldsymbol{x},\sigma_x^2))}{\sqrt{2}K}\right)$$

on the MNIST handwritten digit (LeCun & Cortes, 2010) and CIFAR-10 datasets (Krizhevsky, 2009) for 1-Lipschitz fully connected and convolutional neural networks, respectively.

For comparison, we also implemented the hinge losses with MSCR but without smoothing, i.e., averaging over input noise, $\epsilon_{M,l_2}(\boldsymbol{x},c,\sigma_x^2=0)$ (MCR) and with the certified radius of *Randomized Smoothing* $\epsilon_{01,l_2}\left(\boldsymbol{x},c,\sigma_x^2\right)$ as the lower bound given in Eq. 9 under the Zhen's approximation (we refer to it as Zhen loss). In addition to them, for completeness, we also used the Multiclass Hinge loss (MH) $\ell_{\mathrm{MH}}\left(\boldsymbol{x},c,d\right) = \sum_{i\neq c}\max(0,d - (o_c(\boldsymbol{x}) - o_i(\boldsymbol{x})))$ that tries to maximize the difference from all incorrect ones (Anil et al., 2019; Li et al., 2019b), the Softmax Cross-Entropy (SCE) $\ell_{\mathrm{SCE}}\left(\boldsymbol{x},c\right) = -\log\frac{\exp o_c(\boldsymbol{x})}{\sum_i \exp o_i(\boldsymbol{x})}$ that is the most widely used loss for classifiers but neither enforces margin maximization nor acts on a smoothed classifier, and the Smoothed Classifier Softmax Cross-Entropy (SC-SCE) $\ell_{\mathrm{SC\text{-}SCE}}\left(\boldsymbol{x},c\right) = -\log\frac{\exp \mu_c(\boldsymbol{x})}{\sum_i \exp \mu_i(\boldsymbol{x})}$. We summarize these losses in Table 1.

To keep the networks 1-Lipschitz throughout the learning, we used ReLU activation while keeping the weight matrices orthogonal. For the fully connected network, this means that the rows of the weight matrices are mutually orthogonal (Appendix C.2.1), and for convolutions, filters are mutually and "internally" orthogonal (Appendix C.2.2). "Orthogonal weights" means that all their singular values are 1, which means that they are 1-Lipschitz in the $l_2$-norm sense. We used algorithms that guarantee an approximate orthogonalization of weight matrices for every training step (Björck & Bowie, 1971; Li et al., 2019b).

Further details of the experiments, including network architecture, hyperparameters, robustness metrics, datasets, evaluation of output statistics, as well as a qualitative overview of orthogonal convolutions are in Appendix C.

### 4.2 Hyperparameter search

Figure 2 shows the colormap of test data clean accuracy for MNIST experiments, area under the curve of certified radius curve $AUC_{CR,l_2}$ through the margin (worst-case robustness), and area under the curve of noise accuracy $AUC_{GN}$ (average-case robustness) for different combinations of hyperparameters $d$ and $\sigma_x^2$ on MSCR and Zhen losses.

First, these plots show the well-documented generalization-robustness trade-off. For both losses, we can see that hyperparameter pairs with lower clean accuracy (darker areas in the first column) are inversely

Table 1: Losses used in this work. *Margin-based* are the ones that explicitly try to maximize the margin. *Smoothed* are the ones that use a smoothed classifier. Acronyms are MSCR: Margin Smoothed Certified Radius loss (proposed here), Zhen: Zhen's loss (*Randomized Smoothing*), MCR: Margin Certified Radius loss (MSCR without smoothing), MH: Multiclass Hinge loss, SCE: Softmax Cross-Entropy, and SC-SCE: Smoothed Classifier Softmax Cross-Entropy.

| | Loss | | | | | |
| --- | :---: | :---: | :---: | :---: | :---: | :---: |
| | **MSCR** | **Zhen** | **MCR** | **MH** | **SCE** | **SC-SCE** |
| **Margin-based** | ✓ | ✓ | ✓ | ✓ | ✗ | ✗ |
| **Smoothed** | ✓ | ✓ | ✗ | ✗ | ✗ | ✓ |

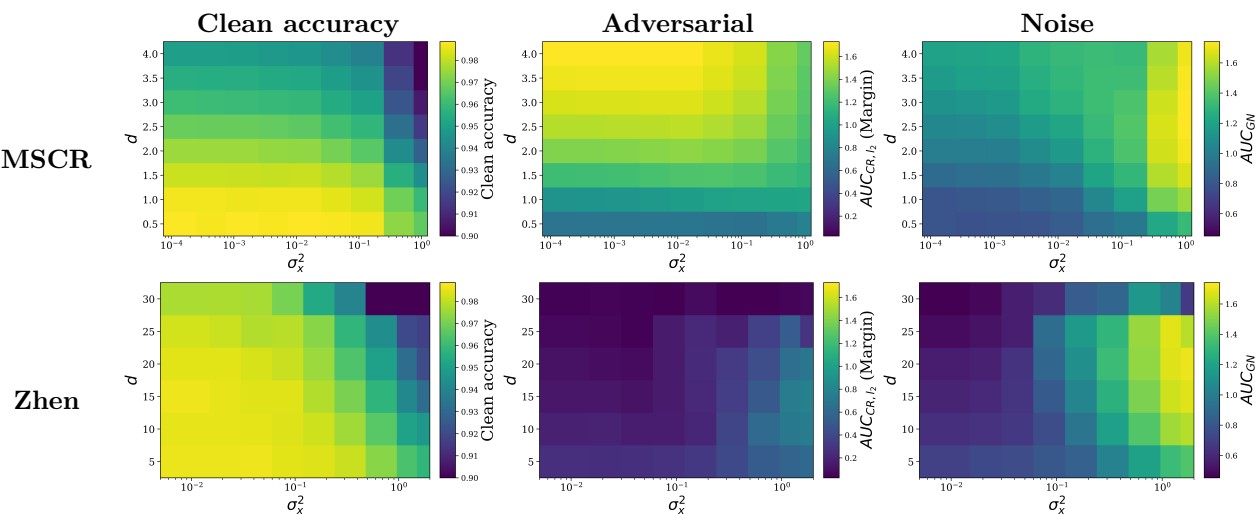

Figure 2: Heatmap of clean accuracy, $AUC_{CR,l_2}$ (margin), and $AUC_{GN}$ for MSCR and Zhen losses. For each column, the colormaps share the same maximum and minimum.

correlated with higher robustness (lighter areas in the second and third columns). Second, we point out that for the MSCR case the two parameters contribute to robustness in complementary ways: while a larger $d$ in general improves the adversarial robustness, by controlling the worst-case distance to the decision boundary, a larger $\sigma_x^2$ induces higher robustness to noise, since a classifier trained in this case learns to correctly classify on average even for stronger random perturbations. On the other hand, for Zhen loss, $\sigma_x^2$ seems to be responsible for tuning the robustness, with little relation to $d$. This dependence on $\sigma_x^2$ can be understood by *Randomized Smoothing* employing only $\sigma_x^2$ to quantify the robustness with no direct enforcement on the non-smoothed worst-case.

In figure 3 we show the colormap of the $AUC_{CR,l_2}$ through *Randomized Smoothing* (Cohen et al., 2019). Interestingly, different from figure 2, in which adversarial robustness with respect to the margin certified radius and noise robustness indicates different kinds of robustness, we see that indeed the *Smoothed Robustness*, as captured by *Randomized Smoothing*, bridges adversarial and noise robustness, with regions of higher robustness in between the regions of higher adversarial and noise robustness.

### 4.3 Robustness comparison

Since a highly robust classifier has lower clean accuracy, and vice versa, to compare the robustness improvement we restrict the parameter pairs with clean accuracy within a range Acc $\pm \Delta$ and select the ones with highest $AUC_{CR,l_2}$ for *Randomized Smoothing* certified radius, since we saw that it nicely merges worst- and average-case. In figures 4 and 5 we show the robustness curves from the robustness metrics described in

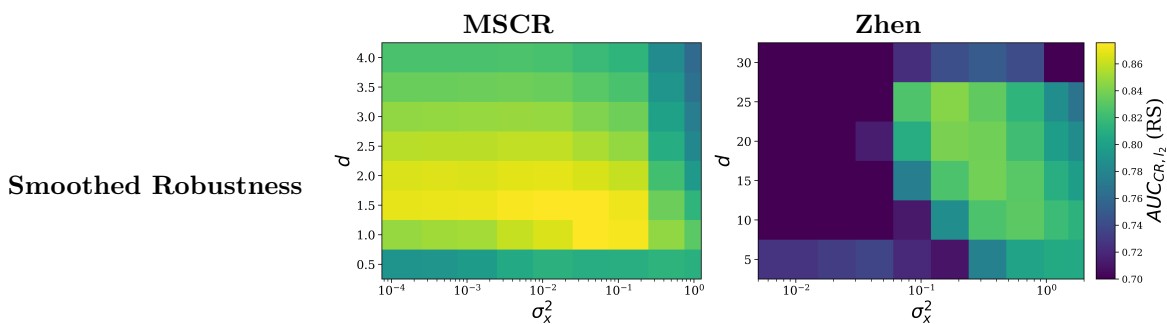

Figure 3: Heatmap of $AUC_{CR,l_2}$ of certified radius obtained through *Randomized Smoothing* (Cohen et al., 2019) for MSCR and Zhen losses. The colormaps share the same maximum and minimum.

Appendix B for conditions $98.5 \pm 0.3$ (i.e., non-clean accuracy degradation case) and $95.5 \pm 0.3$ (i.e., clean accuracy degradation case), respectively, for MNIST dataset.

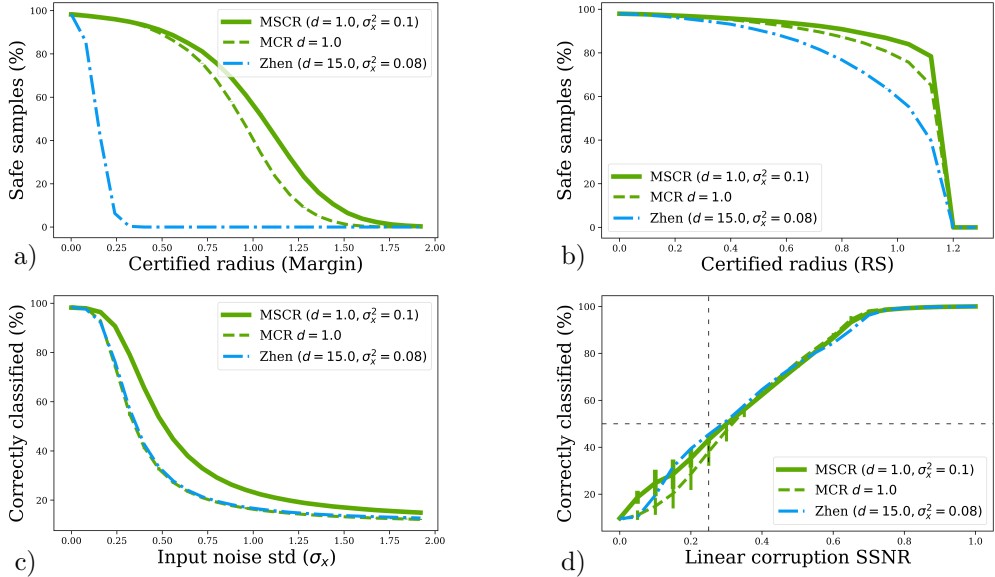

Figure 4: Robustness comparison for MNIST dataset between MSCR, MCR and Zhen losses for the highest $AUC_{CR,l_2}$ for *Randomized Smoothing* within $98.5\pm0.3$ (%) clean accuracy. For (a) and (b) the vertical axis is the amount of safe samples, horizontal axis is the certified radius for margin CR and *Randomized Smoothing* CR, respectively; For (c) and (d) the vertical axis is the accuracy, and horizontal axis the perturbation intensity of Gaussian noise (c) and Linear image corruption (d). The dashed horizontal and vertical lines in (d) indicate the approximate 50% performance drop point in humans from Jang et al. (2021). Each curve is the average of three trainings, and the error bars are shown.

In Figure 4, we see that the MSCR loss achieves the highest adversarial (a and b) and noise (c) robustness and a linear corruption (d) robustness similar to the Zhen loss. Interestingly, while the usual non-smoothed margin loss (MCR) achieves higher adversarial robustness than does the Zhen loss (a and b), it has similar noise robustness (c) and worse linear corruption robustness (d). The main takeaways here are, first, how even the MCR loss is more adversarially robust compared to the Zhen loss, both in terms of the margin and *Randomized Smoothing* certified radius. Second, smoothing the margin significantly improves its noise robustness, as expected, and slightly improves its adversarial robustness. Third, even though the Zhen loss showed lower adversarial and noise robustness than did the MSCR, it showed slightly better linear image corruption robustness, similar to what we would expect human subjects to achieve. These differences

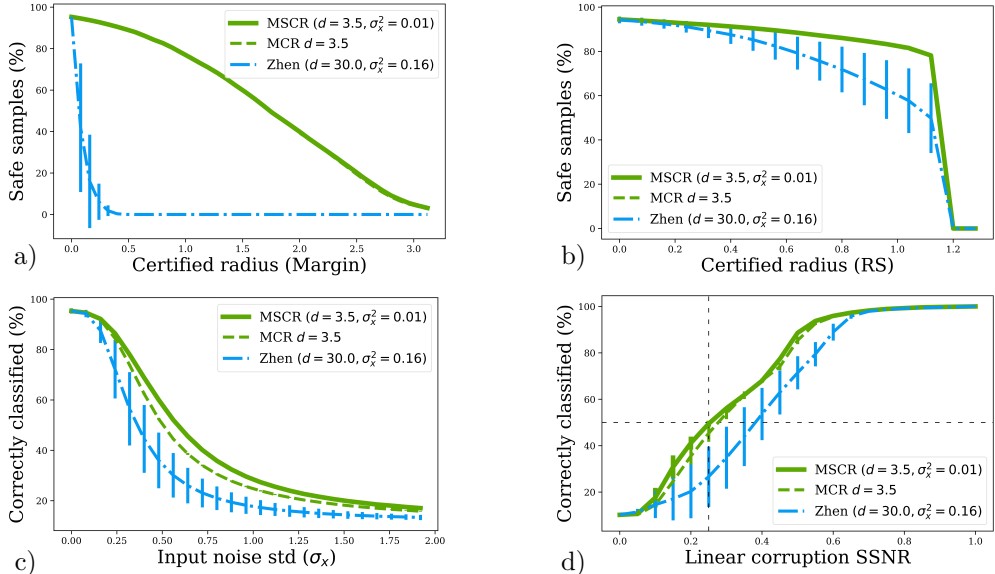

Figure 5: Robustness comparison for MNIST dataset between MSCR, MCR and Zhen losses for the highest $AUC_{CR,l_2}$ for *Randomized Smoothing* within $95.5 \pm 0.3$ (%) clean accuracy. For (a) and (b) the vertical axis is the amount of safe samples, horizontal axis is the certified radius for margin CR and *Randomized Smoothing* CR, respectively; For (c) and (d) the vertical axis is the accuracy, and horizontal axis the perturbation intensity of Gaussian noise (c) and Linear image corruption (d). The dashed horizontal and vertical lines in (d) indicate the approximate 50% performance drop point in humans from Jang et al. (2021). Each curve is the average of three trainings, and the error bars are shown.

demonstrate that evaluating a classifier's robustness is not trivial and that no single metric rules out a classifier as the most robust one.

Now from figure 5, different from the previous case, we observe that MSCR and MCR losses behave more similarly, with an overall slight improvement in robustness for the MSCR loss in the noisy (c) and linear corruption (d) settings. In particular, the Zhen loss showed an improvement in its noise (c) and *Randomized Smoothing* (b) robustness, with the former surpassing the MSCR and MCR losses, at the expense of higher training variance, as shown by the error bars, and decreased margin robustness (a). For the sake of completeness, we show the curves obtained for the other losses as continuous lines in Figure 6.

The robustness obtained with these other losses (given similar clean accuracy) was, for all cases, lower than or similar to that obtained with MSCR (continuous green) or MCR (dashed green) losses. Curiously, we found that the Multiclass Hinge loss (MH), widely used in the robustness literature, yields a performance similar to that of the SCE, which does not enforce any kind of robustness. The SC-SCE loss is a simple alternative to probability-based losses (as in *Randomized Smoothing*); however, we found that, at least under the 1-Lipschitz constraint, there is a large decrease in clean accuracy, with best accuracy of 95.6%, making it an unsuitable choice.

## 4.4 CIFAR-10 experiments

For the MNIST experiments we carried a grid search over all losses' hyperparameters; however, due to computational constraints, we trained NNs on CIFAR-10 only for the MSCR, MCR and Zhen losses. Additionally, instead of a grid search, we found the best combination of hyperparameters by first picking values close to those from the $98.5 \pm 0.3$ results and changing them to neighboring values. Because the architecture used was relatively small (Appendix C) and therefore of low capacity, we noticed that the clean accuracy did not exceed 64%, so we selected the hyperparameters for the best adversarial robustness with a clean accuracy $62.5 \pm 0.2$. The obtained accuracy curves are in figure 7.

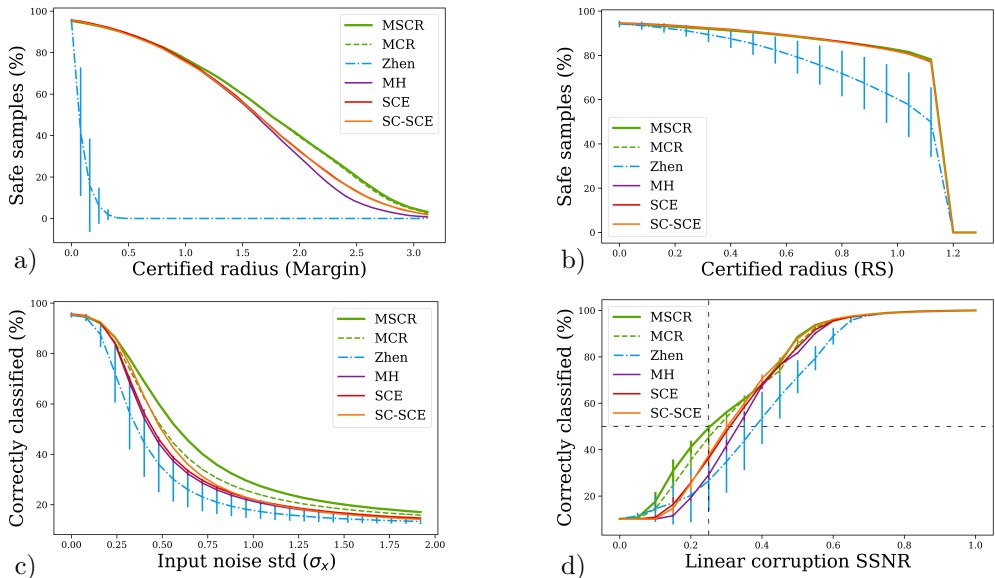

Figure 6: Robustness comparison for between all losses described in Section 4.1 for the highest $AUC_{CR,l_2}$ for *Randomized Smoothing* within $95.5 \pm 0.3$ (%) clean accuracy. For (a) and (b) the vertical axis is the amount of safe samples, horizontal axis is the certified radius for margin CR and *Randomized Smoothing* CR, respectively; For (c) and (d) the vertical axis is the accuracy, and horizontal axis the perturbation intensity of Gaussian noise (c) and Linear image corruption (d). The dashed horizontal and vertical lines in (d) indicate the approximate 50% performance drop point in humans from Jang et al. (2021). Each curve is the average of three trainings, and the error bars are shown.

From the two left plots we see that the curves for MSCR and MCR overlap for the margin certified radius but MSCR improved over MCR for noise robustness. This indicates that while the $d$ hyperparameter contributes to the margin, the $\sigma_x^2$ contributes to noise robustness. And indeed, we see an improvement of MSCR under *Randomized Smoothing* robustness. On the other hand, Zhen loss obtained a much higher noise robustness (c), and a slightly better linear corruption robustness (d), but at the cost of lower adversarial robustness (a) and (b).

# 5 Discussion

Motivated by recent attempts in bridging worst- and average-case robustness analyses of classifiers, in the present work, we employed the smoothed analysis framework of complexity of algorithms to propose the Smoothed Robustness Analysis (SRA) framework, which includes *Randomized Smoothing* as a special case when the 0-1 loss is its performance metric (Sec. 2). Considering the argument from the smoothed analysis literature that the noise variance should be small, which cannot be achieved by the usual *Randomized Smoothing*, we proposed the margin loss as a more suitable performance metric for SRA and derived the Margin Smoothed Certified Radius (MSCR), which is expected to be a more reliable robustness certification as it does not vanish for small noise variances (Sec. 3). To validate the proposal, we conducted experiments to maximize the MSCR and showed improvements in both adversarial and noise robustness (Sec. 4). Here, we discuss the interpretations and limitations of the work and outline possible future directions.

## 5.1 Multiclass hinge losses

We proposed the margin loss defined by the difference between the largest incorrect and correct outputs as a suitable performance metric for SRA. Because the margin provides a simple proxy of the distance to the decision boundary, it has been widely used in the context of margin maximization as a regularization (Tsuzuku et al., 2018; Zhang et al., 2021). Other works proposed the use of a summation of all pairwise dif-

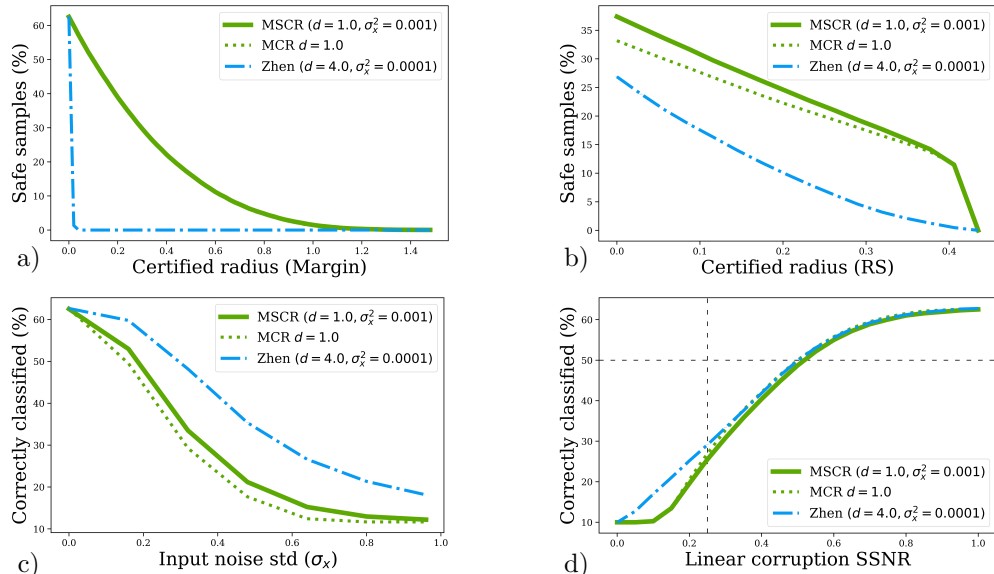

Figure 7: Robustness comparison between MSCR, MCR and Zhen losses for the highest $AUC_{CR,l_2}$ for *Randomized Smoothing* within 62.5±0.2 (%) clean accuracy on CIFAR-10.. For (a) and (b) the vertical axis is the amount of safe samples, horizontal axis is the certified radius for margin CR and *Randomized Smoothing* CR, respectively; For (c) and (d) the vertical axis is the accuracy, and horizontal axis the perturbation intensity of Gaussian noise (c) and Linear image corruption (d). The dashed horizontal and vertical lines in (d) indicate the approximate 50% performance drop point in humans from Jang et al. (2021). Each curve is the average of three trainings, and the error bars are shown.

ferences between incorrect and correct outputs as the loss to improve the robustness of Lipschitz-constrained networks (Anil et al., 2019; Li et al., 2019b). Since, in our experiments, the networks trained with the latter (MH) showed lower robustness than those trained with the former (MCR) (fig. 6), we discuss the reasons for this difference and possible future directions related to the use of other more suitable losses.

Both margin losses are part of a larger group of large margin losses from the support vector machine literature called multiclass hinge losses. Vapnik (1998) and Weston & Watkins (1999) independently proposed extending the binary hinge loss to the multiclass case by taking the sum of pairwise differences between correct and incorrect outputs, in which the MH loss is a special case. Until then, multiclass SVMs were treated as $|C| \cdot (1 - |C|)$ binary SVMs between all outputs, where $|C|$ is the number of classes, which resulted in a larger computational overhead. Crammer & Singer (2001) then proposed a simplified multiclass hinge loss that depends only on the difference between the correct class output and the largest incorrect, i.e., the margin loss in our work.

To better understand the statistical learning properties of these losses, Zhang (2004) showed that these two losses are not Fisher consistent, i.e., in the infinite sample limit, the index of the largest output is not necessarily the index of the correct posterior classification probability of the Bayes classifier. More precisely, the MH loss is consistent if the probability of the correct output is greater than 0.5 or if the probability of the second largest is less than $\frac{1}{|C|}$, and the margin loss is consistent only if the probability of correct classification is greater than 0.5. This result is related to our finding that as long as the smoothed margin is negative, the probability of correct classification is greater than 0.5, but not determined otherwise.

Since this consistency is generally a desirable feature, Liu (2007) proposed using a modification of the margin loss to make it consistent by truncating the margin loss so that it stops increasing at some value. The truncated margin loss can be written as the difference between the usual margin loss and a shifted margin loss $\ell_{TM}(\boldsymbol{x}, c, d) = \ell_M(\boldsymbol{x}, c, d) - \ell_M(\boldsymbol{x}, c, -d) = \mathrm{ReLU}(d + (o_m(\boldsymbol{x}) - o_c(\boldsymbol{x})) - \mathrm{ReLU}(-d + (o_m(\boldsymbol{x}) - o_c(\boldsymbol{x})))$, and it has a maximum of $2d$ when the margin for input $\boldsymbol{x}$ is $o_m(\boldsymbol{x}) - o_c(\boldsymbol{x}) \geq d$. Wu & Liu (2007) used

this loss to improve the accuracy of multiclass SVMs to ignore sample outliers, hence a possible direction is to obtain an approximation of the smoothed truncated margin loss, similar to the one from the present work, and verify whether it helps in the robustness-accuracy trade-off by ignoring few samples that might contribute heavily to the loss.

A recent work by Delattre et al. (2024) improves *Randomized Smoothing* CR by using the Lipschitz constant of the classifier together with the *sparsemax* function. Sparsemax was proposed by Martins et al. (2016) as a (almost everywhere) differentiable embedding on the simplex, like the softmax, but with most values as zero, hence the sparse. They show that for the binary case, sparsemax is the "'hard' sigmoid", i.e., 0 valued, then linear, until it reaches 1. Subtracting this value from 1 for the binary case leads to a truncated hinge loss, similar to the one in the SVM literature. Whether there is a direct connection between them we leave for future investigations.

Here we add that, unlike Dogan et al. (2016), who empirically showed in several datasets that the MH loss, for the case of support vector machines, has better accuracy than other hinge losses, we found the margin (MCR) loss to perform better. Interestingly, we found the MCR loss to significantly improve the overall robustness compared to other usual losses, but also to be unstable if used on unconstrained NNs. Hence, it might be the best cost-performance option for improving robustness in Lipschitz-constrained NNs.

## 5.2 Robustness-accuracy trade-off

In Section 4.2, we showed that, by tuning the hyperparameters $d$ and $\sigma_x^2$, we achieve different degrees of adversarial and noise accuracy, respectively, at the cost of clean accuracy. Although this robustness-accuracy trade-off has been widely demonstrated (e.g. Su et al., 2018; Cohen et al., 2019; Li et al., 2019a), to the best of our knowledge, only Laugros et al. (2019) showed evidence of adversarial and noise robustness corresponding to two different kinds of robustness, i.e., a model robust to AEs is not necessarily robust to random perturbations and vice versa. Using losses with parameters that explicitly account for these two robustness, we provide additional evidence for the findings of Laugros et al. (2019).

We also point out that while early works defended that this trade-off is unavoidable (Tsipras et al., 2019; Zhang et al., 2019), recent accounts favor the idea that this is a consequence of model limitations (Stutz et al., 2019; Olfat & Aswani, 2020; Yang et al., 2020; Leino et al., 2021; Li et al., 2022). More specifically, recent works argue that enforcing a tight global Lipschitz constant (Richardson & Weiss, 2021) and proper loss hyperparameter selection (Béthune et al., 2022) in Lipschitz-constrained networks can suffice to avoid this trade-off. Given the 1-Lipschitzness of the networks in the present work and that methods for exact calculation of Lipschitz constants of feedforward ReLU neural networks are known (Jordan & Dimakis, 2020; Bhowmick et al., 2021), we leave for future work the computation of the constants of the trained networks to verify whether they correlate with the robustness-accuracy gap.

Despite intensive efforts, we still need a definitive solution to the problem of adversarial robustness. Recent works show evidence that top-down information about context and expectation improves general classification robustness (Huang et al., 2020; Alamia et al., 2023), indicating that this might play a central role in achieving the robustness observed in biological brains. From the neuroscience side, a recent work by Guo et al. (2022) showed that robust networks are sensitive to adversarial attacks similar to those measured in primate brains, hypothesizing that an error correction mechanism might be involved. Thus, future works that address this trade-off should take into consideration this possible limitation of feedforward neural networks.

## 5.3 Relationship to 'intermediate-q' robustness

We mentioned how SRA connects to *Randomized Smoothing* in the case of the 01 loss. The work by Rice & Bair (2021) proposes a method to study 'intermediate-q robustness' that bridges these two cases based on the $l_q$-norm of a *smooth* loss $\ell(\boldsymbol{x}, c)$ over a measure $\mu(\boldsymbol{x})$ that depends on input $\boldsymbol{x}$,

$$\|\ell(\boldsymbol{x}, c)\|_{\mu(\boldsymbol{x}), q} = \mathbb{E}_{x' \sim \mu(\boldsymbol{x})}[|\ell(\boldsymbol{x}', c)|^q]^{\frac{1}{q}}. \tag{10}$$

Their method considers the usual definition of losses as nonnegative scalar functions, so $|\ell(\boldsymbol{x}', c)| = \ell(\boldsymbol{x}', c)$. When $q \to \infty$ and $\mu(\boldsymbol{x})$ is a uniform ball, one obtains the worst-case, i.e., the maximal loss within the ball.

When $q = 1$, this expression gives the average loss on the measure $\mu(\boldsymbol{x})$. The connection of SRA with this method is that the averaging step from SRA generates a smooth function $\ell(\boldsymbol{x}, c) = \mathbb{E}_{\tilde{\boldsymbol{x}} \sim \rho(\boldsymbol{x})}[\ell'(\tilde{\boldsymbol{x}}, c)]$, $\ell'(\boldsymbol{x}, c)$ being a loss not necessarily smooth, and then SRA (for nonnegative losses) can be written in this $l_q$-norm approach as

$$\|\ell(\boldsymbol{x}, c)\|_{\Omega(\boldsymbol{x}), l_\infty} = \mathbb{E}_{x' \sim \Omega(\boldsymbol{x})}[\mathbb{E}_{\tilde{\boldsymbol{x}} \sim \rho(\boldsymbol{x}')}[\ell'(\tilde{\boldsymbol{x}}, c)]^\infty]^{\frac{1}{\infty}}. \tag{11}$$

The main difference between SRA and the method of Rice & Bair (2021) is that while the former incorporates the worst and average cases as two axes, one controlled by the size of the worst-case region $\Omega(\boldsymbol{x}) \subset \mathbb{X}$ and the other by the variance of the smoothing distribution $\rho(\boldsymbol{x})$, the latter combines them as two extremes in a spectrum. Additionally, SRA characterizes the robustness of a classifier by stating whether it is "safe" under a choice of noise distribution and adversarial region given some loss. Then, a robustness comparison is done by evaluating which classifier is safe for larger regions (e.g. larger certified radius) and input variances. The approach based on the $l_q$-norm acts as a robustness metric, in which by choosing the norm and measure, in theory, a classifier with larger $l_q$-norm is more robust.

### 5.4 Limitations and other future directions

Given the nature of this work as a first exposition of Smoothed Robustness Analysis, our experiments were limited only to 1-Lipschitz fully connected and convolutional neural networks under isotropic Gaussian noise, dealing with $l_2$-norm radius certification. Future works with margin SRA, therefore, should consider other noise distributions and estimation methods of the smoothed margin that do not require the assumptions of symmetry and independence of outputs, thus allowing its application to NNs other than 1-Lipschitz. A lower bound on the margin certified radius that takes into account the complex relationship between pairwise margins, in line with the soft margin proposed by Fazlyab et al. (2023), is another possible direction.

Although we obtained improvements in noise robustness with MSCR compared to MCR on CIFAR-10 while maintaining its clean accuracy and adversarial robustness, the difference was more subtle than that for MNIST. We hypothesize that for larger CNNs that achieve higher clean accuracy smoothing is more beneficial, since overparametrized models tend to have more complex geometries. Due to time constraints, we leave this an open question.

Finally, since smoothed analysis was originally proposed to capture the algorithmic complexity observed in practice, another possible direction is to test the SRA as a measure of neural network complexity. Intuitively, when comparing multiple classifiers under the same loss given the same input region Omega and distribution rho, the one with the smallest average (over test samples) SRA has a decision boundary that better separates unseen data even under small noise. This might be related to the notion of compression from information bottleneck (Shwartz-Ziv & Tishby, 2017), in which neural networks that correctly classify inputs with more noise better compress their relevant information, resulting in lower model complexity. SRA may contribute to elucidating the connection between robustness and complexity.

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

## A  Proofs and derivations

### A.1  Relation of smoothed margin and classification probability

**Proposition 3.** *Let the output layer $o(\boldsymbol{x})$ of a classifier be continuous and symmetrically distributed, i.e., the marginal distribution of $i$-th dimension $o_i(\boldsymbol{x})$ is symmetrically distributed. If $\mathbb{E}_{\boldsymbol{x}' \sim \rho(\boldsymbol{x}, \boldsymbol{\theta})}[\mathcal{L}_M(\boldsymbol{x}', c)] < 0$, then the probability of correct classification $\mathcal{P}_c(\boldsymbol{x}; \rho(\cdot)) > 0.5$.*

*Proof.* We have the following inequality for the smoothed margin $\bar{\mathcal{L}}_M(\boldsymbol{x}, c; \rho(\cdot))$ of continuous and symmetrically distributed outputs,

$$\bar{\mathcal{L}}_M(\boldsymbol{x}, c; \rho(\cdot)) = \mathbb{E}_{\boldsymbol{x}' \sim \rho(\boldsymbol{x}, \boldsymbol{\theta})}[\max_{i \neq c} o_i(\boldsymbol{x}')] - \mu_c > \text{med}[\max_{i \neq c} o_i(\boldsymbol{x})] - \mu_c = \text{med}[\max_{i \neq c} o_i(\boldsymbol{x})] - \text{med}[o_c(\boldsymbol{x})],$$

where $\mu_c$ is the mean correct output, med$[\cdot]$ refers to the median, the inequality step holds in the case of continuous symmetrically distributed outputs since the distribution of $\max_{i \neq c} o_i(\boldsymbol{x})$ is positively skewed [2], which

---

[2] We refer to Arellano-Valle & Genton (2008); Navarro & Arevalillo (2023) for a detailed account on exact analysis of order statistics

means that $\mathbb{E}_{\boldsymbol{x}' \sim \rho(\boldsymbol{x}, \boldsymbol{\theta})}[\max_{i \neq c} o_i(\boldsymbol{x}')] > \text{med}[\max_{i \neq c} o_i(\boldsymbol{x}')]$, and $\mu_c = \text{med}[o_c(\boldsymbol{x})]$ since $o_c(\boldsymbol{x})$ is symmetrically distributed.

If the difference between the two medians is negative, then the probability of $o_c(\boldsymbol{x}) > \max_{i \neq c} o_i(\boldsymbol{x})$ is greater than 0.5. Therefore, if $\bar{\mathcal{L}}_M(\boldsymbol{x}, c; \rho(\cdot)) < 0$, the probability of correct classification is greater than 0.5. $\qquad\square$

## A.2 Smoothed radius certification with margin function

Here we derive a certified radius of a smoothed $K_p$-Lipschitz classifier based on the margin function. First we show that the margin of such classifier is $2^{\frac{1}{q}} K_p$-Lipschitz, where $\frac{1}{q} + \frac{1}{p} = 1$, then we use the result that the smoothed form of a scalar $K$-Lipschitz function is also $K$-Lipschitz to arrive at the certified radius of a smoothed classifier $\epsilon_{l_p}(\boldsymbol{x}; \rho(\cdot)) = -\frac{\bar{\mathcal{L}}_M(\boldsymbol{x}, c; \rho(\cdot))}{2^{\frac{1}{q}} K_p}$.

**Proposition 5.** *The margin function $\mathcal{L}_M(\boldsymbol{x}, c)$ of a $K_p$-Lipschitz classifier with fixed $c$ is $2^{\frac{1}{q}} K_p$-Lipschitz, where $\frac{1}{q} + \frac{1}{p} = 1$.*

*Proof.* To prove this proposition, we first need to show that the margin as a function of the output $\boldsymbol{o}(\boldsymbol{x})$ is $2^{\frac{1}{q}}$-Lipschitz, then we finish the proof by using the composition of Lipschitz functions. First, the margin $\mathcal{L}_M(\boldsymbol{x}, c) = \boldsymbol{o}_m(\boldsymbol{x}) - \boldsymbol{o}_c(\boldsymbol{x})$ can be written as $\mathcal{L}_M(\boldsymbol{x}, c) = \boldsymbol{v}_{c,m} \cdot \boldsymbol{o}(\boldsymbol{x})$, where $\boldsymbol{v}_{c,m}$ is a vector with 1 at position m, $-1$ at position c, and 0 elsewhere. Then, from the definition of Lipschitz continuity, we need to find an upper bound on $|\mathcal{L}_M(\boldsymbol{x}_1, c) - \mathcal{L}_M(\boldsymbol{x}_2, c)| = |\boldsymbol{v}_{c,m_1} \cdot \boldsymbol{o}(\boldsymbol{x}_1) - \boldsymbol{v}_{c,m_2} \cdot \boldsymbol{o}(\boldsymbol{x}_2)| \, \forall \, \boldsymbol{x}_1, \boldsymbol{x}_2 \in \mathbb{X}$ that is proportional to $\|\boldsymbol{x}_1 - \boldsymbol{x}_2\|_p$.

Then the following inequality holds,

$$
\begin{aligned}
|\mathcal{L}_M(\boldsymbol{x}_1, c) - \mathcal{L}_M(\boldsymbol{x}_2, c)| = |\boldsymbol{v}_{c,m_1} \cdot \boldsymbol{o}(\boldsymbol{x}_1) - \boldsymbol{v}_{c,m_2} \cdot \boldsymbol{o}(\boldsymbol{x}_2)| &\leq |\boldsymbol{v}_{c,m_M} \cdot \boldsymbol{o}(\boldsymbol{x}_1) - \boldsymbol{v}_{c,m_M} \cdot \boldsymbol{o}(\boldsymbol{x}_2)| \\
&= |\boldsymbol{v}_{c,m_M} \cdot (\boldsymbol{o}(\boldsymbol{x}_1) - \boldsymbol{o}(\boldsymbol{x}_2))| \\
&\leq \|\boldsymbol{v}_{c,m_M}\|_q \|\boldsymbol{o}(\boldsymbol{x}_1) - \boldsymbol{o}(\boldsymbol{x}_2)\|_p \\
&= 2^{\frac{1}{q}} \|\boldsymbol{o}(\boldsymbol{x}_1) - \boldsymbol{o}(\boldsymbol{x}_2)\|_p,
\end{aligned}
\tag{12}
$$

where $m_j = \arg\max_{i \neq c} o_i(x_j)$ and $M = \arg\max_{i \in \{1,2\}} v(c, m_i) \cdot \boldsymbol{o}(x_i)$. The first inequality holds since, if $m_M = m_1$, then $v(c, m_1) \cdot \boldsymbol{o}(x1) \geq v(c, m_2) \cdot \boldsymbol{o}(x2) \geq v(c, m_1) \cdot \boldsymbol{o}(x2)$, therefore $|v(c, m_1) \cdot \boldsymbol{o}(x1) - v(c, m_2) \cdot \boldsymbol{o}(x2)| \leq |v(c, m_1) \cdot \boldsymbol{o}(x1) - v(c, m_1) \cdot \boldsymbol{o}(x2)|$ (if $m_M = m_2$ the indices 1 and 2 are exchanged). The second inequality is due to Holder's inequality $|fg| \leq \|f\|_q \|g\|_p$, where $\frac{1}{q} + \frac{1}{p} = 1$. We have the last equality because $\|\boldsymbol{v}_{c,m_M}\|_q = (1^q + |-1|^q)^{\frac{1}{q}} = (2 \cdot 1^q)^{\frac{1}{q}} = 2^{\frac{1}{q}}$.

Using inequality 12 and the fact that the classifier is $K_p$-Lipschitz, i.e., $\|\boldsymbol{o}(\boldsymbol{x}_1) - \boldsymbol{o}(\boldsymbol{x}_2)\|_p \leq K_p \|\boldsymbol{x}_1 - \boldsymbol{x}_2\|_p$, we finally have $|\mathcal{L}_M(\boldsymbol{x}_1, c) - \mathcal{L}_M(\boldsymbol{x}_2, c)| \leq 2^{\frac{1}{q}} K_p \|\boldsymbol{x}_1 - \boldsymbol{x}_2\|_p$. $\qquad\square$

**Corollary 5.1** (Margin-based certified radius). *For a correctly classified input $\boldsymbol{x}$, $\varepsilon = -\frac{\mathcal{L}_M(\boldsymbol{x}, c)}{2^{\frac{1}{q}} K_p}$ is a lower bound on the norm of the minimal perturbation $\boldsymbol{\delta} \in \mathbb{X}$ required to make $\mathcal{L}_M(\boldsymbol{x} + \boldsymbol{\delta}) = 0$. In other words, the classifier correctly classifies $\boldsymbol{x}$ to all perturbations $\boldsymbol{\delta} \in \mathbb{X}$ s.t. $\|\boldsymbol{\delta}\|_p < \varepsilon$.*

To see this, we first note that $\mathcal{L}_M(\boldsymbol{x}, c) < 0$ since $\boldsymbol{x}$ is correctly classified. Second, a minimal successful adversarial perturbation $\boldsymbol{\delta}$ implies $\mathcal{L}_M(\boldsymbol{x} + \boldsymbol{\delta}, c) = 0$. By substituting both in the Lipschitz inequality we have $|\mathcal{L}_M(\boldsymbol{x}, c)| = -\mathcal{L}_M(\boldsymbol{x}, c) \leq 2^{\frac{1}{q}} K_p \|\boldsymbol{\delta}\|_p$, which implies that the minimal successful perturbation has norm $\|\boldsymbol{\delta}\|_p \geq -\frac{\mathcal{L}_M(\boldsymbol{x}, c)}{2^{\frac{1}{q}} K_p}$. Conversely, any perturbation less than that lower bound is guaranteed to not result in misclassification.

To proceed, we need to prove that the smoothed form of a $K$-Lipschitz function due to the application of random noise is also $K$-Lipschitz. For this we use the following definition of smoothed function:

**Definition 1.** *The smoothed form of a function $F : \mathbb{X} \to \mathbb{Y}$ by convolution with a probability density function $\rho(\boldsymbol{x}; \boldsymbol{\theta})$ parametrized by $\boldsymbol{\theta}$ is*

$$\hat{F}\left(\boldsymbol{x}; \boldsymbol{\theta}\right) := \int_{\mathbb{R}^{n_x}} d\boldsymbol{t} \rho(\boldsymbol{t}; \boldsymbol{\theta}) F(\boldsymbol{x} - \boldsymbol{t}).$$

Then we have the following proposition about the lipschitzness of a smoothed scalar function:

**Proposition 6.** *Let $F(\boldsymbol{x})$ be a $K_p$-Lipschitz scalar function of $\boldsymbol{x} \in \mathbb{X}$. Then its smoothed form $\hat{F}(\boldsymbol{x}; \boldsymbol{\theta})$ is $K_p$-Lipschitz, where the subscript $p$ stands for the $l_p$-norm.*

*Proof.*

$$\left| \hat{F}\left(\boldsymbol{x}_1; \boldsymbol{\theta}\right) - \hat{F}\left(\boldsymbol{x}_2; \boldsymbol{\theta}\right) \right| = \left| \int_{\mathbb{R}^{n_x}} d\boldsymbol{t} \rho(\boldsymbol{t}; \boldsymbol{\theta})(F(\boldsymbol{x}_1 + \boldsymbol{t}) - F(\boldsymbol{x}_2 + \boldsymbol{t})) \right|$$

$$\leq \int_{\mathbb{R}^{n_x}} d\boldsymbol{t} \rho(\boldsymbol{t}; \boldsymbol{\theta}) \left| (F(\boldsymbol{x}_1 + \boldsymbol{t}) - F(\boldsymbol{x}_2 + \boldsymbol{t})) \right|$$

$$\leq \int_{\mathbb{R}^{n_x}} d\boldsymbol{t} \rho(\boldsymbol{t}; \boldsymbol{\theta}) K_p \left\| \boldsymbol{x}_1 - \boldsymbol{x}_2 \right\|_p = K_p \left\| \boldsymbol{x}_1 - \boldsymbol{x}_2 \right\|_p,$$

where the first inequality holds because the absolute value of an integral is always less than or equal to the integral of an absolute value, and the second inequality is because $F(x)$ is $K_p$-Lipschitz. $\square$

Finally, using the previous results, we obtain a certified radius of a smoothed classifier based on its smoothed margin.

**Theorem 4.** *Let the* smoothed margin $\hat{\mathcal{L}}_M\left(\boldsymbol{x}, c, \sigma_x^2\right)$ *be defined according to Eq. 4 and Eq. 1. If $\hat{\mathcal{L}}_M\left(\boldsymbol{x}, c, \sigma_x^2\right) < 0$, then $\hat{\mathcal{L}}_M\left(\boldsymbol{x} + \delta, c, \sigma_x^2\right) < 0 \ \forall \ \delta \in \mathbb{R}^{n_x} \ s.t. \ \left\| \delta \right\|_p < -\frac{\hat{\mathcal{L}}_M\left(\boldsymbol{x}, c, \sigma_x^2\right)}{2^{\frac{1}{q}} K_p}$, where $K_p$ is the Lipschitz constant of the classifier and $\frac{1}{q} + \frac{1}{p} = 1$.*

*Proof.* From Proposition 5 we know that the Lipschitz constant of the margin function is $2^{\frac{1}{q}} K_p$. Using Proposition 6, we have that the smoothed margin is also $2^{\frac{1}{q}} K_p$-Lipschitz. Then, using the same argument from Corollary 5.1, we have that the minimal perturbation $\delta \in \mathbb{X}$ s.t. $\hat{\mathcal{L}}_M\left(\boldsymbol{x} + \delta, c, \sigma_x^2\right) = 0$ has norm $\left\| \boldsymbol{\delta} \right\|_p \geq -\frac{\hat{\mathcal{L}}_M\left(\boldsymbol{x}, c, \sigma_x^2\right)}{2^{\frac{1}{q}} K_p}$. Conversely, any perturbation less than that lower bound is guaranteed to not result in misclassification on the smoothed classifier. $\square$

### A.3 Closed-form estimate of smoothed margin

To obtain a closed-form estimate of the smoothed margin, we assume that the outputs follow a multidimensional normal with mean vector $\boldsymbol{\mu}(\boldsymbol{o})$ and covariance matrix $\boldsymbol{\Sigma}(\boldsymbol{o}) = \text{diag}(\mathbf{var}(\boldsymbol{o}))$, where $\text{diag}(\boldsymbol{x})$ denotes the diagonal matrix with the vector $\boldsymbol{x}$ in the diagonal elements and $\mathbf{var}(\boldsymbol{o})$ is the variance vector of the output vector $\boldsymbol{o}$.

For a multivariate normal output $\boldsymbol{o}(\boldsymbol{x})$ we have the smoothed margin,

$$\bar{\mathcal{L}}_M\left(\boldsymbol{x}, c, \sigma_x^2\right) = \mathbb{E}_{\boldsymbol{x} \sim p(\boldsymbol{x})}\left[\max_{i \neq c} o_i\left(\boldsymbol{x}\right) - o_c\left(\boldsymbol{x}\right)\right] = \mathbb{E}_{\boldsymbol{o} \sim p(\boldsymbol{o})}\left[\max_{i \neq c} o_i\left(\boldsymbol{x}\right)\right] - \mu_c$$

$$= \sum_{i \neq c} \int_{-\infty}^{\infty} o_i \rho\left(o_i\right) \prod_{j \neq i} \frac{1}{2}\left[1 + \text{erf}\left(\frac{o_i - \mu_j}{\sqrt{2}\sigma_j}\right)\right] do_i - \mu_c, \tag{13}$$

where in the second line we took the expectation over the output instead of the input, and the indices are sorted in descending order, with 1 being the index of the largest mean among the incorrect outputs.

We approximate the product of CDF to just the CDF of the largest mean among the ones involved in the product, which gives the following expression,

$$\bar{\mathcal{L}}_M\left(\boldsymbol{x}, c, \sigma_x^2\right) \cong \frac{\sigma^2_1}{\sqrt{2\pi\left(\sigma_1{}^2 + \sigma_2{}^2\right)}} \exp\left(-\frac{(\mu_1 - \mu_2)^2}{2\left(\sigma_1{}^2 + \sigma_2{}^2\right)}\right) + \frac{\mu_1}{2}\left(1 + \operatorname{erf}\left(\frac{\mu_1 - \mu_2}{\sqrt{2\left(\sigma_1{}^2 + \sigma_2{}^2\right)}}\right)\right)$$
$$+ \sum_{i=2}^{|C|-1} \frac{\sigma^2_i}{\sqrt{2\pi\left(\sigma_1{}^2 + \sigma_i{}^2\right)}} \exp\left(-\frac{(\mu_1 - \mu_i)^2}{2\left(\sigma_1{}^2 + \sigma_i{}^2\right)}\right) + \frac{\mu_i}{2}\left(1 - \operatorname{erf}\left(\frac{\mu_1 - \mu_i}{\sqrt{2\left(\sigma_1{}^2 + \sigma_i{}^2\right)}}\right)\right) - \mu_c.$$

Empirically, we found that using the terms involving the two largest means is enough to give a close approximation (Appendix D.2). Then, by leaving the two terms, we arrive at the following expression as the smoothed margin,

$$\bar{\mathcal{L}}_M\left(\boldsymbol{x}, c, \sigma_x^2\right) \cong \sqrt{\frac{\sigma_1^2 + \sigma_2^2}{2\pi}} \exp\left(-\frac{(\mu_1 - \mu_2)^2}{2(\sigma_1^2 + \sigma_2^2)}\right) + \mu_2 + \frac{\mu_1 - \mu_2}{2}\left(1 + \operatorname{erf}\left(\frac{\mu_1 - \mu_2}{\sqrt{2(\sigma_1^2 + \sigma_2^2)}}\right)\right) - \mu_c.$$

## A.4 Closed-form estimate of correct class probability

Just like the approximation of the smoothed margin, for the closed-form estimate of correct class probability we make the assumption that the output vector $\boldsymbol{o}(\boldsymbol{x})$ follows a multidimensional normal distribution with independent dimensions. This gives the following integral,

$$\mathcal{P}_c\left(\boldsymbol{x}, \sigma_x^2\right) = \mathcal{P}\left(o_c(\boldsymbol{x}) > o_i(\boldsymbol{x}), \forall i \neq c\right) = \int_{-\infty}^{\infty} \rho\left(o_c\right) \prod_{i \neq c} \frac{1}{2}\left[1 + \operatorname{erf}\left(\frac{o_c - \mu_i}{\sqrt{2}\sigma_i}\right)\right] do_c, \tag{14}$$

By considering only the term of largest mean $\mu_i$ in the product, we have

$$\mathcal{P}_c\left(\boldsymbol{x}, \sigma_x^2\right) \cong \frac{1}{2}\left[1 + \operatorname{erf}\left(\frac{\mu_c - \mu_1}{\sqrt{2(\sigma_c^2 + \sigma_1^2)}}\right)\right].$$

## A.5 ReLU moments

Let the preactivation $s$ of a ReLU unit $h = \operatorname{ReLU}(s)$ follow a Gaussian distribution $\mathcal{N}(\mu, \sigma^2)$. Then, $h$ will follow a rectified Gaussian distribution $\mathcal{N}_{\mathrm{R}}(\mu, \sigma^2)$ with density function

$$\rho(h; \mu, \sigma^2) = \Phi\left(\frac{-\mu}{\sigma}\right)\delta(h) + \frac{\exp(-\frac{(h-\mu)^2}{2\sigma^2})}{\sqrt{2\pi}\sigma}\mathrm{H}(h)$$

where $\Phi\left(x\right) = \frac{1}{2}\left[1 + \operatorname{erf}\left(\frac{x}{\sqrt{2}}\right)\right]$ is the cumulative distribution function of the standard normal distribution; $\delta(x)$ is the Dirac delta function, and $\mathrm{H}(x)$ is the Heaviside step function. The expected value and variance of $h \sim \mathcal{N}_{\mathrm{R}}(\mu, \sigma^2)$ are denoted as follows:

### A.5.1 Expected value of rectified Gaussian distributed variable

The expected value $\bar{h}$ of a unit $h \sim \mathcal{N}_{\mathrm{R}}(\mu, \sigma^2)$ is

$$\bar{h} = \mathbb{E}_{h \sim \mathcal{N}_{\mathrm{R}}(\mu, \sigma^2)}[h] = \int_{-\infty}^{\infty} h\,\rho(h; \mu, \sigma^2)\,dh = \int_0^{\infty} h\frac{\exp(-\frac{(h-\mu)^2}{2\sigma^2})}{\sqrt{2\pi}\sigma}\,dh$$

Substituting $\frac{h-\mu}{\sigma} = z$,

$$\bar{h} = \int_{-\frac{\mu}{\sigma}}^{\infty} (\sigma z + \mu) \frac{\exp(-\frac{z^2}{2})}{\sqrt{2\pi}} \, dz = -\sigma \frac{\exp(-\frac{z^2}{2})}{\sqrt{2\pi}} \Bigg|_{-\frac{\mu}{\sigma}}^{\infty} + \mu \int_{-\frac{\mu}{\sigma}}^{\infty} \frac{\exp(-\frac{z^2}{2})}{\sqrt{2\pi}} \, dz$$

$$= \sigma \frac{\exp(-\frac{\mu^2}{2\sigma^2})}{\sqrt{2\pi}} + \frac{\mu}{2} \left[ 1 + \operatorname{erf}\left( \frac{\mu}{\sqrt{2}\sigma} \right) \right],$$

where the second integral is the complementary cumulative distribution function $1 - \Phi\left( \frac{-\mu}{\sigma} \right)$.

### A.5.2 Variance of rectified Gaussian distributed variable

For the variance $\operatorname{Var}(h) = \mathbb{E}_{h \sim \mathcal{N}_{\mathrm{R}}(\mu,\sigma^2)}[h^2] - \bar{h}^2$, we need the second moment $\mathbb{E}_{h \sim \mathcal{N}_{\mathrm{R}}(\mu,\sigma^2)}[h^2]$,

$$\mathbb{E}_{h \sim \mathcal{N}_{\mathrm{R}}(\mu,\sigma^2)}[h^2] = \int_{-\frac{\mu}{\sigma}}^{\infty} (\sigma z + \mu)^2 \frac{\exp(-\frac{z^2}{2})}{\sqrt{2\pi}} \, dz$$

$$= \int_{-\frac{\mu}{\sigma}}^{\infty} (\sigma^2 z^2 + 2\mu\sigma z + \mu^2) \frac{\exp(-\frac{z^2}{2})}{\sqrt{2\pi}} \, dz$$

$$= \sigma^2 \int_{-\frac{\mu}{\sigma}}^{\infty} z^2 \frac{\exp(-\frac{z^2}{2})}{\sqrt{2\pi}} \, dz + 2\mu\sigma \frac{\exp(-\frac{\mu^2}{2\sigma^2})}{\sqrt{2\pi}} + \frac{\mu^2}{2} \left[ 1 + \operatorname{erf}\left( \frac{\mu}{\sqrt{2}\sigma} \right) \right],$$

obtained substituting $\frac{h-\mu}{\sigma} = z$, and using the previous two integrals of the expected value. The integral with the $z^2$ term is then calculated with integration by parts,

$$\sigma^2 \int_{-\frac{\mu}{\sigma}}^{\infty} z^2 \frac{\exp(-\frac{z^2}{2})}{\sqrt{2\pi}} \, dz = -\sigma^2 z \frac{\exp(-\frac{z^2}{2})}{\sqrt{2\pi}} \Bigg|_{-\frac{\mu}{\sigma}}^{\infty} + \sigma^2 \int_{-\frac{\mu}{\sigma}}^{\infty} \frac{\exp\left(-\frac{z^2}{2}\right)}{\sqrt{2\pi}} \, dz$$

$$= -\mu\sigma \frac{\exp(-\frac{\mu^2}{2\sigma^2})}{\sqrt{2\pi}} + \frac{\sigma^2}{2} \left[ 1 + \operatorname{erf}\left( \frac{\mu}{\sqrt{2}\sigma} \right) \right].$$

Adding the three terms yields the second moment of $h$,

$$\mathbb{E}_{h \sim \mathcal{N}_{\mathrm{R}}(\mu,\sigma^2)}[h^2] = \mu\bar{h} + \frac{\sigma^2}{2} \left[ 1 + \operatorname{erf}\left( \frac{\mu}{\sqrt{2}\sigma} \right) \right],$$

and its variance,

$$\operatorname{Var}(h) = \frac{\sigma^2}{2} \left[ 1 + \operatorname{erf}\left( \frac{\mu}{\sqrt{2}\sigma} \right) \right] - \bar{h}(\bar{h} - \mu).$$

### A.6 CLT for independent rectified Gaussian random variables

In Section 3, to obtain the average classification condition constant for the margin loss we assumed the output layer to be symmetrically distributed, which we show in Appendix D.1 to be empirically satisfied. For wide enough layers this is the case, since for the case of independent, but not necessarily identically distributed, rectified Gaussian random variables, the Central Limit Theorem (CLT) applies, which we prove in this section.

**Proposition 7.** *Let $h_1, \ldots, h_n$ be a sequence of independent rectified Gaussian random variables, where $h_k$ has mean $\bar{h}_k$ and variance $\mathrm{Var}(h_k) = \sigma_k^2$. Then, the sum $\sum_{k=1}^{n} h_k$ satisfies the CLT.*

*Proof.* For the proof we use the following theorem and condition:

**Theorem 8** (Lindeberg theorem (Billingsley, 1995)). *Assume a sequence $x_1, \ldots, x_n$ of $n$ independent random variables, where the random variable $x_k$ has mean $\mathbb{E}[x_k] = 0$ and variance $\sigma_k^2 = \mathbb{E}[x_k^2]$. Then, the sum $\sum_{k=1}^{n} x_k$ converges to a normal distribution $\mathcal{N}(0, s_n^2)$, where $s_n^2 = \sum_{k=1}^{n} \sigma_k^2$, if the Lindeberg condition holds for all $\epsilon > 0$.*

**Condition 2** (Lindeberg condition). *Assume a sequence $x_1, \ldots, x_n$ of $n$ independent random variables, where the random variable $x_k$ has mean $\mathbb{E}[x_k] = 0$ and variance $\sigma_k^2 = \mathbb{E}[x_k^2]$. Then,*

$$\lim_{n \to \infty} \sum_{k=1}^{n} \frac{1}{s_n^2} \int_{|x_k| \geq \epsilon s_n} x_k^2 \, dP_k = 0 \tag{15}$$

*is the Lindeberg condition, where $s_n^2 = \sum_{k=1}^{n} \sigma_k^2$.*

Let us consider a sequence $h_1, \ldots, h_n$ of independent rectified Gaussian random variables, where $h_k$ has mean $\bar{h}_k$ and variance $\mathrm{Var}(h_k) = \sigma_k^2$. Without loss of generality, let $\sigma_1^2 \leq \sigma_2^2 \leq \cdots \leq \sigma_n^2$ and $x_k = h_k - \bar{h}_k$ so that it satisfies[3] $\mathbb{E}[x_k] = 0$. We also define

$$\ell = \arg\max_{k} \left[ \int_{|x_k| \geq \sqrt{n}\epsilon\sigma_1} x_k^2 \, dP_k \right]. \tag{16}$$

Then, for all $\epsilon > 0$,

$$\begin{aligned}
\lim_{n \to \infty} \frac{1}{s_n^2} \sum_{k=1}^{n} \int_{|x_k| \geq \epsilon s_n} x_k^2 \, dP_k &\leq \lim_{n \to \infty} \frac{1}{n\sigma_1^2} \sum_{k=1}^{n} \int_{|x_k| \geq \sqrt{n}\epsilon\sigma_1} x_k^2 \, dP_k \\
&\leq \lim_{n \to \infty} \frac{1}{n\sigma_1^2} n \int_{|x_\ell| \geq \sqrt{n}\epsilon\sigma_1} x_\ell^2 \, dP_\ell \\
&= \frac{1}{\sigma_1^2} \lim_{n \to \infty} \int_{|x_\ell| \geq \sqrt{n}\epsilon\sigma_1} x_\ell^2 \, dP_\ell \\
&= 0,
\end{aligned}$$

where we used $n\sigma_1^2 \leq s_n^2$ for the first and Eq. 16 for the second inequality. The last limit is zero because $P_\ell(x_\ell)$ is a shifted rectified Gaussian distribution, which means that $\lim_{n \to -\infty} P_\ell(x_\ell) = 0$ and $P_\ell(x_\ell)$ rapidly decays for large $x_\ell$, i.e., it decays exponentially while the $x_\ell^2$ part of the integrand grows quadratically.

Therefore, Lindeberg's condition is satisfied, so the sum of independent but not necessarily identically distributed rectified Gaussian random variables follows the CLT. $\qquad\square$

# B   Robustness metrics

Since the motivation for SRA is that adversarial and noise robustness are distinct from each other, we measure the adversarial robustness through the margin certified radius and the noise robustness under isotropic

---

[3]This change of variable shifts the mean of the sum to 0 without changing the variance and the second moment integral of Eq. 15

Gaussian input noise. We also measure the *Smoothed Robustness* through the *Randomized Smoothing* certified radius. For the certified radius, we obtain the "survival" curves for constant increments in the certified radius. For noise robustness, we measured the average accuracy for increments of input variance. For MNIST training, to visualize how both types of robustness changed for different pairs of loss hyperparameters $(d, \sigma_x^2)$ in the MSCR and Zhen losses, we computed the area under the curve $AUC_{CR,l_2}$ of the survival curves of certified radius (Tsai et al., 2021b;a; Li et al., 2024), and similarly propose the area under the curve $AUC_{GN}$ of the noise accuracy curve. The AUC is a simple and intuitive metric of overall robustness since $AUC_{CR,l_p}$ is the average certified radius under $l_p$ attacks, and $AUC_{GN}$ is the expected standard deviation that causes the input to be misclassified on average[4].

While the previous three metrics give information about both worst-, average-case and *smoothed* robustness, they do not tell how robust a classifier is in real-world scenarios, or how it compares to human subjects. Motivated by this, we implemented the linear image corruption method from Jang et al. (2021). In their work, they were interested in how robust Deep NNs are compared to humans in image classification and proposed a simple method of corrupting an image by building an image that is a linear combination of the original times a constant $\Gamma \in [0, 1]$ and a noise image times $(1 - \Gamma)$. With this method, we can not only compare the different trained NNs but also qualitatively determine how far they are from human robustness, since one of the results from Jang et al. (2021) is that human subjects in general reached a 50% decrease in accuracy when the original image was about 25% of the final corrupted image.

## B.1 Adversarial robustness

To assess the worst-case robustness, i.e., adversarial robustness, we computed the certified radius. The first is through the deterministic margin from Eq. 6, the second through *Randomized Smoothing* with the CERTIFY algorithm by Cohen et al. (2019). For the latter, we used $10^5$ samples, $\alpha = 0.001$, and $\sigma = 0.25$. After obtaining the certified radius for each correctly classified sample we obtained the certified radius curve as the total number of samples with CR greater than a specific radius. From these curves we obtained their area under the curve $AUC_{CR,l_2}$ as

$$\text{AUC}_{CR,l_2}(\mathbb{X}_{test}) = 0.5\Delta\varepsilon \sum_{j=0} \left(\text{TotCR}(\mathbb{X}_{test}, (j+1) \cdot \Delta\varepsilon) + \text{TotCR}(\mathbb{X}_{test}, j \cdot \Delta\varepsilon)\right),$$

where the summation is over the integer $j$ and $\text{TotCR}(\mathbb{X}_{test}, j \cdot \Delta\varepsilon)$ is the total number of CR greater than radius $j \cdot \Delta\varepsilon$.

## B.2 Noise robustness

To evaluate the average-case robustness, i.e., robustness to noise, we measured how the performance is gradually affected by increasing the standard deviation of an additive isotropic Gaussian noise. For each trained NN the following routine was carried to measure its noise robustness:

1. Measure the noise accuracy for zero variance (i.e., the clean accuracy) $\text{NoiseAcc}(\mathbb{X}_{test}, \sigma_x = 0)$ of all test samples $\mathbb{X}_{test}$

2. Increase the input standard deviation $\sigma_x$ by $\Delta\sigma_x = 0.2$ for MNIST, or $0.2/2$ for CIFAR-10

3. For each input, sample 100 noise vectors[5] $\delta$ from $\mathcal{N}(0, \sigma_x^2 \boldsymbol{I}_{n_x})$, where $\boldsymbol{I}_{n_x}$ is the identity matrix with $n_x$ diagonal elements, and $n_x$ is the input dimension

4. Calculate the accuracy for each of the noise samples and store its average in $\text{NoiseAcc}(\mathbb{X}_{test}, \sigma_x)$

5. Repeat from step 2 until the average noise accuracy is less than 0.13 ($\frac{1}{|C|}$ plus a small cutoff value, in this case 0.03)

---

[4]Since for large input variances the noise accuracy converges to one divided by the number of classes, we resort to normalization of the noise accuracy curve described in B.2

[5]On Appendix D.3 we discuss how we chose this number of samples

To quantify the robustness to noise we also employ the area under the curve $\text{AUC}_{GN}$. While for the survival curve of the certified radius there is a big enough radius that drops the accuracy to zero, yielding a finite AUC, for the noise accuracy the accuracy drops to around $1/|C|$, where $|C|$ is the number of classes, so we cannot apply the AUC directly. Because of this, we normalize the noise accuracy so that it maintains the same value for zero noise, but decreases the lowest accuracy from $1/|C|$ to 0, which gives a finite AUC.

Then, we have the normalized noise accuracy $\text{NoiseAcc}'(\mathbb{X}_{test}, \sigma_{x,j})$ with $\sigma_{x,j} = j \cdot \Delta\sigma_x$,

$$\text{NoiseAcc}'(\mathbb{X}_{test}, \sigma_{x,j}) = \frac{(1 - \alpha_j)\text{NoiseAcc}(\mathbb{X}_{test}, \sigma_{x,j})}{1 - \alpha_j\text{NoiseAcc}(\mathbb{X}_{test}, \sigma_{x,j})}$$

where $\alpha_j$ is a $[0,1]$ mapping, being 0 for $j = 0$ and 1 for $j \to \infty$,

$$\alpha_j = 1 - \frac{\text{NoiseAcc}(\mathbb{X}_{test}, j \cdot \Delta\sigma_x) - \min_k \text{NoiseAcc}(\mathbb{X}_{test}, k \cdot \Delta\sigma_x)}{\text{NoiseAcc}(\mathbb{X}_{test}, 0) - \min_k \text{NoiseAcc}(\mathbb{X}_{test}, k \cdot \Delta\sigma_x)}.$$

The meaning of this expression is that for $\sigma_x = 0$ the accuracy remains the same, while the minimal accuracy becomes 0 for $\sigma_x \gg 0$.

Then, using this normalized accuracy we can calculate the $\text{AUC}_{GN}$ as

$$\text{AUC}_{GN} = 0.5\Delta\sigma_x \sum_{j=0} \left(\text{NoiseAcc}'(\mathbb{X}_{test}, (j+1) \cdot \Delta\sigma_x) + \text{NoiseAcc}'(\mathbb{X}_{test}, j \cdot \Delta\sigma_x)\right).$$

The values of $\Delta\varepsilon$ and $\Delta\sigma_x$ were picked in such a way that it takes around 10 of these increments to completely degrade the performance.

### B.3 Linear corruption robustness

The linear image corruption from Jang et al. (2021) allows a simple way of controlling the quality of an image by adjust a single parameter $\alpha \in [0,1]$, where for $\alpha = 1$ we have the original image and for $\alpha = 0$ there is a complete loss of information. In their method, given an image $\boldsymbol{x} \in [0, 255]^{n_x}$ and a "gray" noisy image $\boldsymbol{n}$, i.e., all pixels valued with $255/2$ plus a small additive noise $\delta \sim \mathcal{N}(0, 6/255)$, the final image to be presented is a linear combination

$$\boldsymbol{I} = \alpha\boldsymbol{x} + (1 - \alpha)\boldsymbol{n}$$

where the parameter $\alpha \in [0,1]$ can be interpreted as the signal-to-signal-plus-noise ratio, which when equals to 1 there is a pure signal and when 0 there is pure noise. Then, by presenting images with different values of $\alpha$ in 0.05 increments, they measured the reduction in classification performance in both human and NN classifiers.

## C   Complete experimental settings

### C.1   NN Architecture

For the MNIST experiments we used a multilayer perceptron with three hidden layers of 512 units each and ReLU activation, which due to its computational simplicity allowed for an extensive evaluation of all losses considered in 4.1, their respective hyperparameters grid search, as well as the variations in their robustness performance. [6]

For the CNN architecture, we used four convolutional layers of output channel sizes $(8, 8, 16, 16)$, kernel sizes $(4, 3, 4, 3)$ and strides $(2, 1, 2, 1)$, with a single fully connected layer of 512 units and ReLU activations in all

---

[6]The code for all our experiments can be found at https://github.com/ThomRC/sra

layers. We adopted other architecture elements from Li et al. (2019b), such as not using pooling layers, since these are not norm-preserving, and in our case would result in a heavily skewed preactivation distribution, therefore not allowing the use of the moments of the ReLU. Like them, we also use "invertible downsampling" to reduce the dimensionality, since striding is not norm-preserving. So, to have an operation equivalent to stride 2 one reshapes the previous layer by stacking each entry of a $2 \times 2$ non-overlapping patch, which multiplies the channel size by 4, hence decreasing the dimension of the posterior layer while preserving the norm.

## C.2 ReLU moment propagation

In the estimates of the smoothed margin and probability of correct classification, we assumed independent normally distributed outputs. Orthogonal weights are required to mitigate the dependence of preactivations, which for wide layers are approximately normal with mean $\mu_l = \boldsymbol{W}_l \bar{\boldsymbol{h}}_{l-1}$ and covariance matrix $\boldsymbol{\Sigma}_l = [\boldsymbol{W}_l \odot \boldsymbol{W}_l] \operatorname{Var}(\boldsymbol{h}_{l-1})$, where $\boldsymbol{W}_l$ is the weight tensor of l-th layer (i.e., the weight matrix and the convolutional kernel for the fully connected and convolutional cases, respectively), $\odot$ is the elementwise product, $\bar{\boldsymbol{h}}_{l-1}$ is the mean of the (l-1)-th hidden layer and $\operatorname{Var}(\boldsymbol{h}_{l-1})$ its variance, both of same shape as $\boldsymbol{h}_{l-1}$. In training with smoothed losses we implemented moment propagation (Wu et al., 2019) under ReLU activation functions.

Moment propagation consists of computing the mean vector $\mu_l$ and covariance matrix $\Sigma_l$ of the preactivations $\boldsymbol{s}_l$ of the $l$-th layer given by the mean activation of the previous hidden layer $\bar{h}_{l-1}$ and covariance $\operatorname{cov} h_{l-1}$, which in turn are estimated using $\mu_{l-1}$ and $\Sigma_{l-1}$. Although ReLU activation with normally distributed preactivation has simple closed forms for its mean and variance[7], the covariance does not have a closed form expression, thus requiring an approximation (Wu et al., 2019). Fortunately, by using row-orthogonal weight matrices, which mitigate covariances, we do not need to worry about the covariances and need to compute only the mean and variance of ReLU layers.

### C.2.1 Orthogonal weight matrices

We adopted the Björk orthogonalization algorithm (Björck & Bowie, 1971) to obtain orthogonal weight matrices. In the *Björk orthogonalization* there are two parameters, the order of the approximation $p$ and the number of iterations $q$. By setting the order $p = 1$ and starting with a matrix $\boldsymbol{A}_0 = \boldsymbol{A}$ the algorithm computes an approximate orthogonal matrix iteratively as

$$\boldsymbol{A}_{k+1} = \boldsymbol{A}_k \left( \frac{3}{2} I - \frac{1}{2} \boldsymbol{A}_k^\top \boldsymbol{A}_k \right).$$

This sequence of operations, even though resulting in large overhead for more iterations, is amenable to backpropagation, where the entries of the 0-th matrix $\boldsymbol{A}$ are updated, and easily implemented.

For the implementation of *Björk orthogonalization* we used the code from Anil et al. (2019) as base. However, different from their work, in which a progressive orthogonalization is acceptable, i.e., increasing the iteration number to better approximate an orthogonal matrix as the training converges, and similar to Berg et al. (2019), where they had to enforce orthogonality for their normalizing flows approach, we start the training with 30 iterations and every 10 epochs the layerwise number of iterations $q_l$ is decreased to the minimum that satisfies $\frac{\sum_{i \neq j} \| \tilde{\boldsymbol{W}}_{l,i} \cdot \tilde{\boldsymbol{W}}_{l,j}^\top \|}{n_l * (n_l - 1)} \leq \varepsilon$, i.e., the mean pairwise dot product between different rows of the weight matrix from $l$-th layer obtained after the orthogonalization, $\tilde{\boldsymbol{W}}_l$, should be less than $\varepsilon$, $n_l$ being the number of units in the $l$-th layer. We found that in general this mean is not less than $10^{-8}$, even for a large number of iterations, so we set $\varepsilon = 10^{-7}$ in all experiments.

### C.2.2 Orthogonal convolutional layers

There exist several algorithms for building orthogonal convolutional layers (Singla et al., 2022; Singla & Feizi, 2021; Yu et al., 2022; Xu et al., 2022; Trockman & Kolter, 2021; Meunier et al., 2022). We used the implementation of block convolution orthogonal parameterization from Li et al. (2019b) since it uses the *Björck*

---

[7]We derive both in the Appendix A.5.

*orthogonalization* as backbone, which we already had implemented. We show their algorithm in 1, with some minor modifications for conciseness, where $c_i$ and $c_o$ are the layer's number of input and output channels, $k$ is the kernel size, $\square$ is the block convolution[8] operator defined as $[\boldsymbol{A}\square\boldsymbol{B}]_{i,j} = \sum_{i'=\infty}^{\infty} \sum_{j'=\infty}^{\infty} [A_{i',j'} B_{i-i',j-j'}]$, where $\boldsymbol{A}$ and $\boldsymbol{B}$ are $m_A \times n_A$ and $m_B \times n_B$ block matrices, $\boldsymbol{A}_{i,j} = 0$ for $i \not\subset \{0, m_A\} \vee j \not\subset \{0, n_A\}$. To generate symmetric projectors $\boldsymbol{P}$ and $\boldsymbol{Q}$ with rank $\lfloor \frac{c_i}{2} \rfloor$, Li et al. (2019b) used orthogonal square matrices $\boldsymbol{M}_i$ and $\boldsymbol{N}_i$ and multiplied each row by a "masking" vector with the first $\lfloor \frac{c_i}{2} \rfloor$ entries equals to one and the others to zero.

---

**Algorithm 1:** Block Convolution Orthogonal Parametrization

**Input:** $c_o \times c_i$ orthogonal matrix $\boldsymbol{H}(\boldsymbol{H}_0)$, $c_i \times \lfloor \frac{c_i}{2} \rfloor$ orthogonal matrices $\boldsymbol{M}_i(\boldsymbol{M}_{i,0})$, $\boldsymbol{N}_i(\boldsymbol{N}_{i,0})$ for $i$ from 1 to $k-1$, assuming $c_i \geq c_o$

**Result:** Orthogonal Convolution Kernel $\boldsymbol{W} \in \mathbb{R}^{kc_o \times kc_i}$

Initialize $\boldsymbol{W}$ as a $1 \times 1$ convolution with $\boldsymbol{W}[0,0] = \boldsymbol{H}$;

**for** *i from 1 to k-1* **do**

$\quad \boldsymbol{P}, \boldsymbol{Q} \leftarrow \boldsymbol{M}_i \boldsymbol{M}_i^\top, \boldsymbol{N}_i \boldsymbol{N}_i^\top$ ; $\quad\quad\quad\quad\quad$ ▷ Construct half-rank symmetric projectors;

$\quad W \leftarrow W \square \begin{bmatrix} \boldsymbol{PQ} & \boldsymbol{P}(\boldsymbol{I} - \boldsymbol{Q}) \\ (\boldsymbol{I} - \boldsymbol{P})\boldsymbol{Q} & (\boldsymbol{I} - \boldsymbol{P})(\boldsymbol{I} - \boldsymbol{Q}) \end{bmatrix}$;

**end**

**Output:** $\boldsymbol{W}$

---

This algorithm is a modification of the one proposed by Xiao et al. (2018) to allow the parameterization of the orthogonal matrices with unconstrained ones $\boldsymbol{H}_0$, $\boldsymbol{M}_{i,0}$ and $\boldsymbol{N}_{i,0}$ that will have its values updated during training. However, unlike these works in that the goals of Xiao et al. (2018) and Li et al. (2019b) were to preserve the input and gradient norms, respectively, our reason for using orthogonal convolutions is to mitigate the covariance between units that emerges from the non-orthogonality.

To illustrate how non-orthogonality can be an issue in our case, let us consider the case of filters of size $2 \times 2 \times c_{in}$ and stride 1. The first pixel in the first channel of the convolutional layer is the inner product of this filter and the first $2 \times 2 \times c_{in}$ pixels in the input. The second pixel of the first channel is the product between the filter and the $2 \times 2 \times c_{in}$ pixels in the input after shifting one pixel to the right. Because there is an overlap of $2 \times 1 \times c_{in}$ input pixels, the covariance between these two pixels from the second layer is zero only if the first and second $2 \times 1 \times c_{in}$ parts of the filter are orthogonal. Hence, non-orthogonal convolutions can create strong correlations between units, even with isotropic input noise, thus hindering the use of our closed-form estimates of moments of ReLU and smoothed losses.

To gain some insight into why this final block matrix is equivalent to an orthogonal convolutional layer, here we describe it qualitatively (for a rigorous and quantitative description, we refer the reader to (Kautsky & Turcajov, 1994) and Li et al. (2019b)). The algorithm by Xiao et al. (2018) is an extension to the 2D case of the one from Kautsky & Turcajov (1994), which in turn shows how to build 1D discrete orthonormal wavelets. In terms of wavelet analysis (Kautsky & Turcajov, 1994; Chui, 1992), the output matrix $\boldsymbol{W}$ is an orthonormal 2D discrete wavelet, and the $r$-th rows of each submatrix $\boldsymbol{W}_{i,j}$ correspond to the coefficients of one element of the orthonormal basis of the wavelet. By stacking each of these $k^2$ rows in the order of the indices $i$ and $j$, we have one filter of depth $c_i$ and size $k$.

For a wavelet to be orthonormal it needs to have block autocorrelation - the "non-flipped" convolution with itself - as the zero matrix for all values except at "zero", i.e., when the wavelet "overlaps" itself. To roughly sketch why this is the case, we first point out that all elements in its first column end with the projector $\boldsymbol{Q}_{k-1}$, while all elements in the last column end in $\boldsymbol{I} - \boldsymbol{Q}_{k-1}$. When calculating the autocorrelation of this matrix, all the products of the first column entries with the transpose of the entries of the last columns result in the zero matrix, since $\boldsymbol{Q}_{k-1}$ and $\boldsymbol{I} - \boldsymbol{Q}_{k-1}$ are orthogonal. Then, due to sequential convolutions with $2 \times 2$ block matrices, the columns from 2 to $k-1$ have both entries that end in $\boldsymbol{Q}_{k-1}$ and $\boldsymbol{I} - \boldsymbol{Q}_{k-1}$, but the columns $i$ s.t. $1 < i \leq k/2$ always have the same number of $\boldsymbol{Q}_{k-1}$ entries as the column $k-i$ has entries

---

[8]The intuition of the block convolution is that, just like the convolution between two functions "flips" the function over each axis, it moves the "flipped" block matrix on the left over the block matrix on the right, and each entry of the resulting block matrix is the sum of the matrix multiplication of the overlapping entries.

$I - Q_{k-1}$. This kind of anti-symmetry in the distribution of projectors in $W$ due to its generation from block convolutions of projectors, with directly neighboring elements having products of projections differing in only one projector, results in the autocorrelation of the matrix $W$ having all entries $0$, except the center one, that is the autocorrelation of the unshifted matrix.

### C.3 Datasets

The MNIST dataset (LeCun & Cortes, 2010) consists of 60000 training and 10000 test $28 \times 28$, and CIFAR-10 (Krizhevsky, 2009) of 50000 training and 10000 test $32 \times 32 \times 3$ 8-bit images, labeled from 0 to 9 with the number reflecting the respective class of the image. The only pre-processing was the scaling of each pixel from the range 0-255 to 0-1 by dividing all input elements by 255.

### C.4 Loss hyperparameters search

We used $\sigma_x^2 \in \{0.001, 0.005, 0.01, 0.05, 0.1\}$ for the SC-SCE loss. For the MCR and MH losses, we used $d \in \{0.5, 1.0, 1.5, 2.0, 2.5, 3.0, 3.5, 4.0\}$. For the Zhen loss we did a grid search over $\sigma_x^2 \in \{0.01, 0.02, 0.04, 0.08, 0.16, 0.32, 0.64, 1.28, 1.75\}$ and $d \in \{5, 10, 15, 20, 25, 30\}$. For the MSCR, over $\sigma_x^2 \in \{10^{-4}, 5 \times 10^{-4}, 10^{-3}, 5 \times 10^{-3}, 10^{-2}, 5 \times 10^{-2}, 10^{-1}, 5 \times 10^{-1}, 10^0\}$ and $d \in \{0.5, 1.0, 1.5, 2.0, 2.5, 3.0, 3.5, 4.0\}$.

### C.5 Training

The weight update algorithm was Adam (Kingma & Ba, 2017) with a learning rate of 0.01 and 0.005 for the MNIST and CIFAR-10 experiments, respectively, with the usual hyperparameters $\beta_1 = 0.9$, $\beta_2 = 0.999$, starting the weight updates only after 2000 steps of updating the adaptive parameters $m$ and $v$ for learning stability.

Finally, we found that using cosine annealing as proposed by Loshchilov & Hutter (2017) with warm restarts resulted in generally better and faster convergences. We implemented the method described in their appendix, so we will not go into details, but the method works by decaying the scale of the learning rate from 1 to 0 in the first $n_0$ epochs according to the cosine function and, after reaching the $n_0$-th epoch, the scaling goes back to 1 and starts the decay again. When the scaling goes back to 1 this is called a warm restart and helps avoid poor local minima and, to improve stability, the decay takes more time in-between warm restarts, with $n_{k+1} = m \cdot n_k$ epochs from the $k$-th to the $(k + 1)$-th warm restart, where $m$ is a constant integer greater than 1. By choosing $n_0 = 20$ and $m = 2$, in epoch 300 it finishes the third warm restart, with 160 epochs, which we found to result in stable training.

## D Complementary experimental results

### D.1 Approximate output normality

In Section 4.1, we assumed that the output layer is distributed according to a multivariate normal distribution with independent outputs to obtain closed-form estimates of the smoothed margin and the probability of correct classification. Here we compare the values of the empirical smoothed margin and probability of correct classification with the ones obtained through numerical integration to show that for the case of orthogonal weight matrices with wide enough layers (512 units) this assumption is fairly satisfied.

We carried the numerical integration of the integral equations 13 and 14 used for the estimates with `integrate.quad` function from python's `scipy` package. The integration interval was selected to be $\mu_j \pm 4\sigma_j$, where $j$ is the index of the output variable that the integration is taken, since smaller values would result in larger integration errors, while larger ones would increase the computational cost with small improvements in precision.

For the empirical evaluation, we obtained sample estimates of the probability of correct classification,

$$\mathcal{P}_c'\left(\boldsymbol{x}, \sigma_x^2\right) = \frac{\sum_{i=1}^N \mathbf{1}_{o_c(\boldsymbol{x}_i) > o_j(\boldsymbol{x}_i), \forall j \neq c}}{N},$$

where $N = 10^6$ was the number of samples taken by forwarding different samples of input perturbation from an isotropic Gaussian noise.

Both the sample and the numerical integration values were obtained by increments of $\Delta\sigma_x = 0.032$ in the input standard deviation, using a NN with settings from Section C.5 after 30 training epochs. In figure 8 we see that the approximate output normality assumption is satisfied, given that the numerical and empirical values agree well.

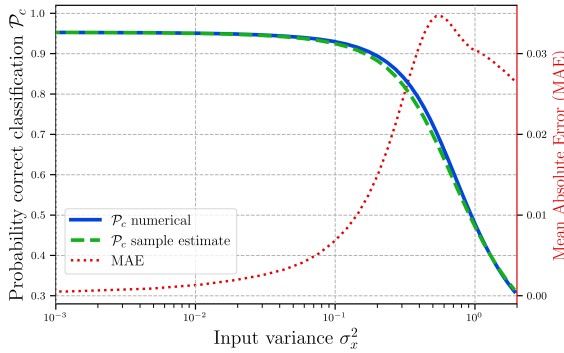

Figure 8: Comparison of correct classification probability $\mathcal{P}_c$ through numerical integration (continuous blue) and sample estimate (dashed green) for different values of input variance $\sigma_x^2$. The second axis shows the mean absolute error between the numerical and sample estimate values (dotted red). Each curve corresponds to the average over test samples after 30 epochs.

## D.2 Closed-form approximations

In figure 9 we present how close the approximations (dashed purple lines) of the smoothed margin (Eq. 7) and probability of correct classification (Eq. 8) are, respectively, to their integral equations 13 and 14 (continuous blue lines). We can see that, overall, the approximations are close to their numerical values, starting to diverge only for larger values of input variance. We also point out that the maximal mean absolute error occurs when half the samples are misclassified on average, i.e., when $\mathcal{P}_c = 0.5$ and $\mathcal{L}_M = 0$ on average.

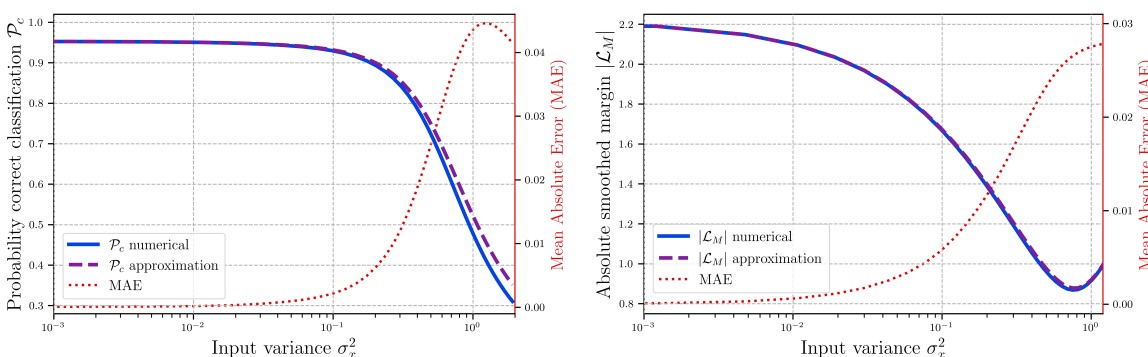

Figure 9: Comparison of approximation (purple) and numerical integration (blue) values of probability of correct classification (left) and absolute smoothed margin (right) for increasing input variance. The second axis shows the mean absolute error between the numerical and approximation values (red line). Each curve corresponds to the average over test samples after 30 epochs, and we plot the absolute smoothed margin for better visualization, since as the variance increases, the signs of the SM start to change making the average SM close to zero.

### D.3 Accuracy under noise

In Appendix B.2 we describe using 100 samples for the estimate of accuracy under noise. In figure 10 we show that as the number of samples increases, the variance of the noise accuracy estimate decreases. For this plot we used input variance of $\sigma_x = 1$, which is a large variance. For 100 samples, the error bar becomes close to 1 in 10000, and since 10000 is the size of our test sample we used this value.

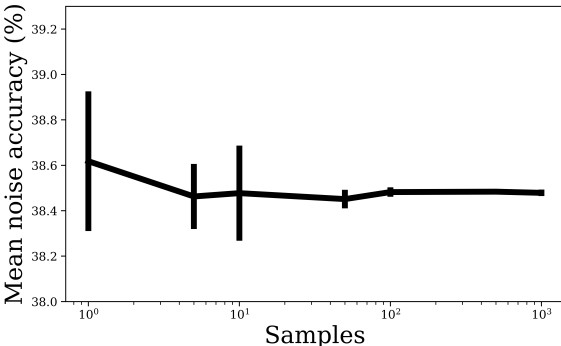

Figure 10: Plot of the estimate noise accuracy for different number of samples. The error bars were computed over five estimates using the shown sample number.

