# OpenReview forum: "Smoothed Robustness Analysis: Bridging worst- and average-case robustness analyses via smoothed analysis"
_TMLR — Accepted by TMLR_

### Review · Reviewer_uodL · 2023-11-07

**Summary Of Contributions:**

This paper proposes a new framework for analyzing the robustness of classifiers, called Smoothed Robustness Analysis (SRA), which combines worst-case and average-case analyses to provide a more comprehensive understanding of classifier robustness. In addition to proposing a new framework for analyzing classifier robustness, the paper also introduces a new margin loss-based function called Margin-based Cross-Entropy Robustness (MCR) loss. This loss function is designed to enhance robustness under the Smoothed Robustness Analysis (SRA) framework. The paper includes experimental results demonstrating the effectiveness of the MCR loss function on MNIST. Therefore, the contributions of this paper are twofold: first, it introduces a new framework for analyzing classifier robustness that bridges the gap between worst-case and average-case analyses, and second, it proposes a new margin loss-based function that enhances robustness under the SRA framework.

**Audience:**

Yes

**Claims And Evidence:**

No

**Requested Changes:**

**Critical**
* Conducting experiments on a more complex dataset
* Conducting experiments on other architectures (outside Lipschitz-constrained MLPs)
* Conducting experiments against state-of-the-art methods with standard robustness metrics

**Would strengthen**
* Reducing the paper length
* Reducing the number of acronyms
* More cleanly separating the approximations from the exact theoretical results

**Strengths And Weaknesses:**

**Strengths**

The particular way the authors unify worst and average case analysis is novel, and the authors make an interesting connection to complexity theory. Also, the theoretical analysis in the paper is solid. The background sections of the paper (1 and 2) are quite comprehensive and detailed, and the methods are also carefully explained. Finally, the empirical results are nicely presented.

**Weaknessess**

The paper has several areas for improvement in my view:

Empirical results
* Experiments are only conducted on MNIST; I would encourage the authors to at least experiment on CIFAR-10 or a dataset of similar complexity to improve the paper's significance and strength of claims
* It's unclear whether the robustness metrics introduced by the authors such as ASR and AMS can be compared to prior work. I would recommend using metrics proposed in prior literature when applicable.
* Experiments are limited to Lipschitz-constrained NNs. This makes it difficult to make claims about the relative efficacy of the authors' proposed method to prior robustness methods which may not be limited to this architecture.
* Similarly, experiments are only conducted on MLP networks. It would be ideal to consider CNN architectures as well, especially given that the authors consider an image dataset. This can help improve the empirical results.
* Ideally, the authors can produce state-of-the-art results for smoothed robustness; this requires comparing the proposed method against state-of-the-art methods on well-performing architectures under standard robustness metrics.

Clarity
* The paper is quite long (over 12 pages), which makes it difficult to grasp the key ideas and results in the paper quickly. I recommend reducing the length.
* The authors frequently use acronyms to the point of confusion; indeed, a full page in the appendix is devoted to all the acronyms introduced. I would suggest reducing the number of acronyms if possible.
* It would be good if the authors could better separate the theoretical assumptions and approximations (e.g. Eqn 12) from exact results.

---

> ### Author Response · Authors · 2024-02-26
> **Major improvements in the manuscript and extended experiments**
>
> Dear Reviewer,
>
> We are grateful to the reviewer for the valuable comments. We have conducted additional simulations and largely revised the manuscript to make it as short as possible and simultaneously to explain the points of the proposed idea as clearly as possible without any verbose or wordy expressions. Below, we list our point-by-point responses. We believe that the revisions address all of the concerns raised by the reviewer.
>
> * Experiments only on MNIST and MLP networks
>
> Thank you for the valuable comments. Following the comments, we carried additional experiments on CIFAR-10 with the Convolutional Neural Network. We implemented the block convolution orthogonal parameterization (BCOP) algorithm from Li et al. (2019). While we used a relatively small CNN and could not perform a grid search due to time constraints, the results showed that both margin losses improved the adversarial robustness and interestingly the smoothed margin led to an improvement in noise robustness while maintaining the clean accuracy and adversarial robustness. We added a new section (Section 4.4) to the revised manuscript with mentioning these results in Abstract and Introduction.
>
> * I would recommend using metrics proposed in prior literature when applicable.
>
> Thank you for the comment. Considering this and also the fact that PGD is not a very strong attack, instead of the metrics we have used in the previous manuscript, in the revised manuscript, we showed the results using the adversarial robustness by measuring the margin certified radius and the randomized smoothing certified radius of a prior work (Cohen et al. (2019)). Then, for the colormaps we used the Area Under the Curve (AUC) of the cumulative certified radius curves, which has been used in prior works. Also, for the noise robustness, following the second reviewer’s comment, we used more samples to obtain the average noise accuracy, and measured, similarly, the AUC of the noise accuracy curve. We thank the reviewer for bringing up the point.
>
> * More cleanly separating the approximations from the exact theoretical results
> * Reducing the paper length and number of acronyms
>
> We are really sorry for our previous manuscript, which was unnecessarily long and difficult to follow. We extensively revised it to clearly explain the main idea and logic of our proposal, removing unnecessary references, and reducing the acronyms. We believe that the readability of the manuscript has improved now, as it clearly explains our logic. In the revision, we specified the prerequisites of each statement to distinguish theoretical and approximated results. For length, while we added a page to explain CIFAR-10 results, the pages are shortened by almost half (from about 14 to 7 pages) before the section on numerical simulations, and even the main text is reduced from 20 to 15 pages. Acronyms are also limited to ‘Smoothed Robustness Analysis (SRA)’ and ‘Margin Smoothed Certified Radius (MSCR),’ which are our main proposals, except for the results section where we keep some acronyms for ease of figure plotting and comparison. (We summarize them in the caption of Table 1 for ease of reference.)
>
> * Experiments are limited to Lipschitz-constrained NNs
>
> We used Lipschitz-constrained NNs with orthogonal layers for our experiments because (i) they allow easy computation of the certified radius, which we need to use the margin as a loss, (ii) with layer orthogonality, we can mitigate the covariances and apply the CLT to the output layer to obtain closed-form estimates of the smoothed certified radius, and (iii) the stability of 1-Lipschitz NN allowed us to use the margin maximization loss directly rather than a regularizer, which helped us observe their effects directly. We agree that it would be helpful to alleviate this constraint, but for the first exposition of our framework, we wanted to start from the basics. We would also like to comment that various works have been provided very recently that efficiently improve scalability and reduce computational costs for Lipschitz networks. We state this limitation in Sec. 5.4 with mentioning the recent progress of Lipschitz NN Sec.1.1.3.
>
> * Produce state-of-the-art results for smoothed robustness
> * Conducting experiments against state-of-the-art methods with standard robustness metrics
>
> We agree that it must be important to use it to guide state-of-the-art approaches. However, just like our reply to the previous comments, in this paper, we wanted to focus only on the exposition of the Smoothed Robustness Analysis with experiments to demonstrate how the framework can be used to guide a practical choice, namely the use of the margin in the loss to improve the robustness. To clearly show this, we emphasized in the revised manuscript that our experiments are Proof of Concepts of our proposal in the Introduction and Result section.
>
> We thank the reviewer again for the comment.

---

### Review · Reviewer_augP · 2024-01-19

**Summary Of Contributions:**

The authors propose a view on average- and worst-case robustness analysis based on smoothed analysis (from time complexity analysis of algorithms) and term this view smoothed robustness analysis. The key difference to prior work is the combination of worst- and average-case by taking the maximum (i.e., worst-case) of a smoothed (i.e., average-case) loss. They consider both margin and 0-1 losses. Using an assumption that output logits/probabilities are independently normal distributed and using orthogonal weights to obtain 1-Lipschitz networks. Experiments on MNIST show that this improves robustness over standard large margin or smoothed losses.

**Audience:**

Yes

**Broader Impact Concerns:**

No concerns.

**Claims And Evidence:**

No

**Requested Changes:**

- Address questions on paper and method limitations as outlined above; convincing experimental setup is particularly critical.
- Address concerns about writing/length.

**Strengths And Weaknesses:**

Strengths:
- Interesting combination of randomized smoothing and large-margin/Lipschitz based approaches for certified robustness.
- The smoothed analysis inspired formulation of worst- and average-case robustness could be an impactful direction in defining more reasonable composite threat models for research
- Comprehensive experiments for average- and worst-case robustness on MNIST and clear description of robustness evaluation.
- Comparison to large-margin-only and randomized-smoothing-only losses.

Weaknesses:
- The paper is way too long; and the length is not distributed favorly. Introduction, related work, experimental setup are far too verbose, while the main part is difficult to follow due to the many abbreviations and unclear correspondence between abbreviations and equations.
- In the abstract + intro, the relationship/jump from randomized smoothing to large-margin approaches is also unclear, creating a bit of a disconnect while reading.
- The related work does not do a good job in pointing me to actually relevant and recent papers. There seem to be papers on the connection of large-margin and randomized smoothing [a] and there have been improved losses for large-margin methods (though not using 1-Lipschitz netoworks). [b] still gives a good overview in the related work section.

[a] https://arxiv.org/pdf/2309.16883.pdf
[b] https://arxiv.org/pdf/2309.06166.pdf

- Which equation is actually the proposed MSCR? Equation (12)?
- Assuming outputs are independently normally distributed seems like a strong assumption. The authors argue that this stems from the orthogonal weight matrices. But I am still missing a convincing experiment or reference.
- The experimental setup is not convincing in my opinion. This has several reasons:
-- The authors combine two methods for certified robustness, namely randomized smoothing and Lipschitz-based methods But evaluation does not include any certified evaluation (either following the randomized smoothing way which is not fully certified but with enough samples comes close enough, or Lipschitz-based which is fully certified but requires estimating the Lipschitz constant).
-- The PGD evaluation only considers 5 restarts and PGD is fairly outdated on its own compared to AutoAttack or more recent ensemble attacks. Using only 5 samples for random noise also seems limiting.
- Regarding the method, I am not sure if it is very appealing beyond MNIST. The 1-Lipschitz requirement might not scale to larger networks, thereby defeating the advantage of randomized smoothing which is model-independent. So it combines the worst of both worlds (hard to scale, e.g., [b] only recently managed results on TinyImageNet with Lipschitz methods and specific architectures) and the test-time computational overhead of randomized smoothing.

Conclusion:
I am currently not convinced that this paper is a compelling read for TMLR readers. This is not only because the mentioned writing aspects, but also because of method limitations and ambiguities as well as limited evaluation.

---

> ### Author Response · Authors · 2024-02-26
> **Major improvements in the writing and methodology**
>
> Dear Reviewer,
>
> We are grateful to the reviewer for the constructive comments, and after sincerely considering them, we conducted additional experiments on CIFAR-10 dataset with a CNN, increased the sample size from 5 to 100, and changed evaluation metrics for robustness. We also revised the manuscript entirely to make it as short as possible but simultaneously to clarify its logical framework while carefully explaining the main points of the proposal. We believe that the readability of the revised manuscript is largely improved. Below, we list our point-by-point responses. We hope that they adequately address all of the concerns raised by the reviewer.
>
> * The paper is way too long and verbose, with many abbreviations
> * Unclear relationship/jump from randomized smoothing to large-margin approaches
>
> In the new version, we have revised it to explain it clearly, removing verbose expressions and abbreviations. We believe the readability is now largely improved. To better connect smoothing and margin, we put sentences to explain their relationship in the introduction. For length, even after the additional added content, the main text was reduced from 20 to 15 pages
>
> * The related work does not point me to relevant and recent papers
>
> We appreciate the pointed recent works. We have checked them and related papers and found that these are helpful to strengthen the manuscript. Due to them, we believe that we could considerably improve the section “Lipschitz-constrained NNs” in related works (on page 3) and Discussion about randomized smoothing
>
> * Which equation is MSCR? Equation (12)?
>
> Again, we apologize for the ambiguity. We tried to clarify each of our proposals in the revised manuscript. In the revised manuscript, Eq.(5) gives MSCR, and Eq.(7), which corresponds to Eq.(12) in the previous manuscript, provides the approximation of the margin loss smoothed by the input noise. By putting Eq.(7) to Eq.(5), one can obtain an approximation of MSCR.
>
> * Assuming outputs are independently normally distributed seems like a strong assumption
>
> Considering this, we updated our proof in Appendix A.1 to the one that does not require independence of the outputs. We hope this change improves the practicality of the average case condition of the smoothed margin. Our new proof uses the positive skewness of the distribution for the maximum of random vectors. Please see A.1 for details.
>
> * Weakness of experimental setup -- Neither randomized smoothing nor margin certifications -- PGD is fairly outdated -- Using only 5 samples for noise accuracy estimate
>
> Thank you for the valuable comments. About evaluation metrics, in the revised manuscript, we evaluated adversarial robustness through both margin and randomized smoothing certified radius instead of through PGD. Interestingly, we found an even clearer relationship between the loss hyperparameters $d$ and $\sigma_x^2$ and the margin certified radius and noise robustness (colormap plots). Additionally, in the colormap of randomized smoothing, we can also see that the region with best smoothed robustness lies in between what we found for the margin certified radius and noise robustness (Figure 2 and Figure 3). We also measured sample size dependency of sample estimate of the noise accuracy (Appendix, Figure 10). For a large input variance of $\sigma_x = 1$, we found that around 100 samples guarantees good approximation. In the revised manuscript, thus, we increase the number of samples from 5 to 100 and replot all figures in the main text
>
> * Not sure if appealing beyond MNIST. It combines the worst of both worlds, hard to scale and computational overhead
>
> We sincerely considered the points. First, we performed additional experiments using CIFAR-10 on a CNN. Although we could only use a relatively small CNN due to time constraints, the results showed the validity of the method even beyond MLP with MNIST. We added the results as a new section. About the 1-Lipschitz requirement, we used it for allowing cheap certified radius and training stabilization. While we agree that it would be helpful to alleviate this constraint, for the first exposition of our framework, we wanted to start from the basics. We would also like to comment that various works have recently improved scalability and reduced the computational cost for Lipschitz networks. We sincerely state this limitation in Sec. 5.4 and explain the recent progress of Lipschitz NN in Sec.1.1.3. We also would like to comment that, as pointed out by Delattre et al. (2024), knowing the Lipschitz constant we can bound the variance of the output, which means that as the input noise strength $\sigma$ decreases the sample estimate requires fewer samples for the same confidence interval, while randomized smoothing is famous for its computational cost at smaller variances.
>
> We thank the reviewer again for pointing us to this reference.

---

### Review · Reviewer_qcjM · 2024-02-12

**Summary Of Contributions:**

The paper presents a novel framework for assessing classifier robustness, drawing inspiration from smoothed algorithmic complexity analysis to reconcile worst- and average-case scenarios. It proposes a margin loss-based method that provides a certified radius unaffected by input noise variance, thereby improving robustness. Experimental validation on 1-Lipschitz neural networks using the MNIST dataset shows that this approach enhances both adversarial and noise robustness compared to randomized smoothing, with a goal of nearing human-level performance.

**Audience:**

Yes

**Claims And Evidence:**

Yes

**Requested Changes:**

I think the writing in the paper can be improved. The paper has a lot of technical discussion which may be easier for the reader to read if written slightly better.

- All theorems/propositions and proofs are in the appendix. It would be better for readers if at least statements of the theorems or proofs were in the main paper. Provide concise statements of theorems or proofs in the main paper, with detailed proofs in the appendix.
- Consider integrating lemma/thm statements within discussions, particularly in technical sections like 2.2, to highlight key conclusions.
- Change the section 3 title to something more meaningful than "methods"
- Only in section 2, subsubsections have section numbers (e.g., 2.2.1) but other sections do not. Make this consistent
- Section 3.4, "The loss used for the attack is the 6-th loss evaluated by Carlini & Wagner", what's the 6-th loss?
- Section 3.1, "The computation of smoothed margin isn't trivial, since it involves a multi-dimensional integral over unknown distribution", is this non-trivial or not possible?

**Strengths And Weaknesses:**

Strengths:

- The work introduces a novel framework that is well-motivated by smoothed analysis in algorithmic complexity
- The margin loss-based robustness analysis offers a certified radius that doesn't scale with input noise variance and improves the robustness against adversarial and random noise attacks, which is empirically validated on MNIST
- The paper nicely connects its proposed approach to existing research in RS literature

Weaknesses:

- The experimental evaluation is limited to 1-Lipschitz MLP with ReLU activations on the MNIST dataset and primarily focuses on l2-norm radius certification, limiting the generalizability to other architectures, datasets, and noise distributions. The paper mentions that CIFAR10 evaluation is left for future work, but I'm unsure why experimenting with additional datasets would be so difficult to be left for future work
- The paper relies on assumptions of symmetry and independence of outputs for estimating the smoothed margin, which may not always hold true in real-world scenarios

---

> ### Author Response · Authors · 2024-02-26
> **Major improvements in manuscript writing and correction of inconsistencies**
>
> Dear Reviewer,
>
> We are grateful to the reviewer for the positive evaluations and valuable comments for the manuscript. We conducted additional numerical simulations on CIFAR-10 with the Convolutional Neural Network. We also revised the manuscript almost entirely to improve its writing style. We believe that the readability of the revised manuscript is largely improved now, as it clearly explains our logic. Below, we list our point-by-point responses. We expect that they address all of the concerns raised by the reviewer.
>
> * The experimental evaluation is limited to 1-Lipschitz MLP with ReLU activations on the MNIST dataset and primarily focuses on l2-norm radius certification, limiting the generalizability to other architectures, datasets, and noise distributions. The paper mentions that CIFAR10 evaluation is left for future work, but I’m unsure why experimenting with additional datasets would be so difficult to be left for future work.
>
> Thank you for the constructive comment. We conducted additional experiments on CIFAR-10 with the Convolutional Neural Networks. We implemented the block convolution orthogonal parameterization (BCOP) algorithm from Li et al. (2019). While we used a relatively small CNN and could not perform a grid search due to time constraints, the results showed that both margin losses improved the adversarial robustness and interestingly the smoothed margin led to an improvement in noise robustness while maintaining the clean accuracy and adversarial robustness. We added a new section (Section 4.4) to the revised manuscript and put sentences to mention them in Abstract and Introduction. For Lipschitz constraints, we used this because it allows easy computation of the certified radius and stabilization of the learning. We clearly state these points in the last sentence of Section 1.1.3 in the revised manuscript
>
> * The paper relies on assumptions of symmetry and independence of outputs for estimating the smoothed margin, which may not always hold true in real-world scenarios
>
> Thank you for the comment. Considering this, we have updated our proof in Appendix A.1 to the one that does not require independence of the outputs. We hope this change improves the practicality of the average case condition of the smoothed margin. Our new proof uses the positive skewness of the distribution for the maximum of random vectors. Please see A.1 for details. For the closed-form estimation of the smoothed margin, we agree that the assumptions might be strong. While we have not implemented yet, the use of a relationship between pairwise margins can be a possible direction. We added the comment about this to the last line of the first paragraph of section 5.4.
>
> * All theorems/propositions and proofs are in the appendix. It would be better for readers if at least statements of the theorems or proofs were in the main paper. Provide concise statements of theorems or proofs in the main paper, with detailed proofs in the appendix.
> * Consider integrating lemma/thm statements within discussions, particularly in technical sections like 2.2, to highlight key conclusions.
>
> Thank you so much for the helpful comments. In the revised manuscript, we stated pairs of a theorem and a sketch of its proof in the main text, then provided full proofs in the Appendix as needed. We also revised almost the entire manuscript, including section 2.1, corresponding to 2.2 in the previous manuscript, to clarify its logical framework and to integrate theorems into the discussion.
>
> * Change the section 3 title to something more meaningful than “methods”
> * Only in section 2, subsubsections have section numbers (e.g., 2.2.1) but other sections do not. Make this consistent
>
> Thank you for the comments. Because structure of the previous manuscript was confusing, we reorganize sections in the manuscript. In the revised manuscript, all subsections have their section numbers, and previous section 3 is divided into the latter half of section 3.1 whose title is “Margin Loss” and the first part of section 4 whose title is “Proof of Concept”.
>
> * Section 3.4, “The loss used for the attack is the 6-th loss evaluated by Carlini & Wagner”, what’s the 6-th loss?
>
> We are sorry for the confusing description. It just meant the loss f_6(x) of the seven losses from f_1(x) to f_7(x) that were proposed in their paper. However, we did not use this in the revised manuscript because we have changed our metrics to different ones to respond to the comment raised by another reviewer.
>
> * Section 3.1, “The computation of smoothed margin isn’t trivial, since it involves a multi-dimensional integral over unknown distribution”, is this non-trivial or not possible?
>
> We are sorry for the confusing sentence. The sentence is removed from the revised manuscript.
>
> We thank the reviewer again for the comments.

---

### Decision · Action_Editor_jXMd · 2024-03-25

**Recommendation:** Accept with minor revision

**Comment:**

The authors propose the concept of smoothed analysis to bridge worst case and average case robustness of classifiers and derive theoretical connections between the smoothed analysis proposed and randomized smoothing approaches for certifiable robustness against adversarial examples. They provide both theoretical and empirical justification for this work. Hence, I recommend that the paper be accepted.

**Audience:**

The audience of TMLR working on adversarial robustness of deep learning would find the paper interesting

**Claims And Evidence:**

The definition of smoothed analysis is precisely given and the implications on connecting randomized smoothing, and worst case and average case robustness of classifiers is clearly discussed.

The method is validated empirically on MNIST and CIFAR-10